# Bootstrapping mixed MN correlators in 3D

Stefanos R. Kousvos[1,2,3] and Andreas Stergiou[4,5]

**1** Department of Physics, University of Crete, Heraklion GR-70013, Greece
**2** Institute of Theoretical and Computational Physics (ITCP), Department of Physics
**3** Department of Physics, University of Pisa and INFN, Largo Pontecorvo 3, I-56127 Pisa, Italy
**4** Theoretical Division, MS B285, Los Alamos National Laboratory,
Los Alamos, NM 87545, USA
**5** Department of Mathematics, King's College London,
Strand, London WC2R 2LS, United Kingdom

## Abstract

The recent emergence of the modern conformal bootstrap method for the study of conformal field theories (CFTs) has enabled the revisiting of old problems in classical critical phenomena described by three-dimensional CFTs. The study of such CFTs with $O(m)^n \rtimes S_n$ global symmetry, also known as MN models, is pursued in this work. Systems of mixed correlators involving scalar operators in two different representations of the global symmetry group are considered. Isolated allowed regions are found in parameter space for various values of $m$ and $n$. These "islands" can be separated into two qualitative groups: those close to the unitarity bound and those further away. As a by-product of our analysis generic tensor structures required to bootstrap any $G^n \rtimes S_n$ theory with $G$ arbitrary are worked out.

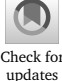

# 1 Introduction

Since its modern revival, the numerical conformal bootstrap [1] has provided a powerful and efficient tool for the study of conformal field theories (CFTs).[1] Its most remarkable successes have been found in $d = 3$ spacetime dimensions, where it has been used to determine critical exponents and other quantities of interest in the $O(N)$ family of models, [4] ($N = 1$), [5] ($N = 2$) and [6] ($N = 3$), as well as the supersymmetric Ising ($N = 1$) variant [7, 8].

Beyond these widely studied examples, it is desirable to extend applications to less examined ones. Three-dimensional multi-scalar theories provide ideal candidates, since they are simultaneously some of the simplest non-trivial theories one may write down, but at the same time they are also physically relevant due to their applications to classical critical phenomena. If one wishes to scan over all the low lying scalar operators as externals in the theory (which would presumably make results more constraining), the number of sum rules one needs to consider is (much) larger than the $O(N)$ case. This is due to the fact that subgroups of $O(N)$ always have (the same or) more irreducible representations than $O(N)$ in the relevant operator product expansions (OPEs).[2] Additionally, one of the most important assumptions in the $O(N)$ case, namely $\Delta_{\phi'} \geqslant 3$ with $\phi'$ the next-to-leading operator in the vector representation of $O(N)$, cannot be used since in a typical multi-scalar theory the next-to-leading operator

---

[1]For a review see [2] and for a pedagogic introduction see [3].

[2]Recently there has been some notable progress regarding techniques for higher dimensional scans in parameter space; see e.g. [9]. See also [10] for an example with two externals but many exchanged parameters scanned over.

in the defining representation has a strongly relevant dimension. This is because the vector representation "$\phi^3$" operator in a generic multi-scalar theory cannot be neglected due to the equation of motion as in the $O(N)$ case, which in turn limits the size of the gaps we can impose in this sector.

There are many three-dimensional multi-scalar models to consider. Perhaps the most well-known one is the cubic model, which is a theory whose global symmetry group consists of the cubic subgroup $\mathbb{Z}_2^3 \rtimes S_3$ of $O(3)$. Conformal field theories with cubic and the more general hypercubic symmetry have been bootstrapped in [11–14]. Many other examples to be considered include fixed points already discovered with the $\varepsilon$ expansion a long time ago,[3] as well as a wealth of new $\varepsilon$ expansion fixed points presented in [16]. With the addition of fermions there are presumably many fixed points to study with the bootstrap, including supersymmetric ones [17].

In this work we continue the study of scalar MN models by considering a multi-correlator study of CFTs with $O(m)^n \rtimes S_n$ global symmetry in three dimensions. These models have been analyzed with the $\varepsilon$ expansion [15, 18–23] as well as the numerical conformal bootstrap [24], where a single-correlator study was pursued. Our goal is to find the minimal set of conditions that allow us to isolate allowed regions in parameter space at the positions of kinks of the single-correlator bounds obtained in [24]. As was explored in [24] and also in [25], bounds of $m = 2$ theories contain two kinks with potential applications to critical phenomena for low $n$ and of theoretical interest at larger $n$. There is no proof that a kink of the type we find in a numerical bootstrap bound must be due to the presence of an actual CFT. However, we will consider such kinks to be strong indicators for the presence of actual CFTs. For an in-depth discussion of experimental realizations of the CFTs we will study and their significance within field theory we refer the reader to [24].

By considering systems of correlators involving operators in multiple irreducible representations (irreps) of the global symmetry group, we manage to find isolated allowed regions (islands) at the positions of kinks of single-correlator bounds. To get these islands we make assumptions on the operator spectrum, but we try to keep the number of such assumptions to a minimum. We find that we typically have to require the presence of a conserved current and stress-energy tensor and gaps to the next-to-leading spin-1 and spin-2 operators in the corresponding irreps. Also, if we wish to obtain an island in the $\Delta_\phi$-$\Delta_S$ plane,[4] we need to require that the leading operator in a specific irrep (called $X$ below) saturates the bound obtained from the single-correlator bootstrap. However, to obtain an island in the $\Delta_\phi$-$\Delta_X$ plane, no such assumption is required. We note that the $\Delta_X$ bound is of interest since it displays pronounced features/kinks in parameter space.

One particularly interesting island is found for the second kink of the $MN_{20,2}$ theory (see Fig. 11 below). This island is obtained in the $\Delta_\phi$-$\Delta_S$ plane, where $\phi$ is the defining and $S$ the singlet irrep of $MN_{20,2}$. A notable feature of this island is that the dimensions $\Delta_\phi$ and $\Delta_S$ that define it are low, while the assumptions we make to obtain it force the dimensions of next-to-leading operators in a variety of irreps to be high.

For the experimentally relevant $MN_{2,2}$ and $MN_{2,3}$ theories we find islands at the first kinks of the corresponding single-correlator bootstrap bounds. One would expect the values of $\Delta_\phi$ and $\Delta_S$ in these islands to match the results obtained in the $\varepsilon$ expansion. This is not the case, however, and the potential resolutions are either that the $\varepsilon$ expansion results in the $\varepsilon \to 1$ limit are not trustworthy or that the bootstrap kink is due to a theory distinct from that of the $\varepsilon$ expansion. We note that at large $m$ and large $n$ the bootstrap results we obtain for $MN_{m,n}$ theories agree very well with the $\varepsilon$ expansion.

---

[3]See for example [15] for a relatively recent review of known fixed points in the $\varepsilon$ expansion below four dimensions.

[4]Where $\phi$ is the order parameter field and $S$ the scalar singlet.

For various values of $m$ and $n$ we initiate a preliminary study of symmetric tensors $Z_{ij}^{ab}$ of $MN_{m,n}$, similar in spirit to [26]. We find a family of kinks that in the large $n$ limit converge to the point $\Delta_Z = \frac{1}{2}$ and $\Delta_Y = 2$, which hints towards a standard Hubbard–Stratonovich description in this limit. Conversely, in the large $m$ limit, the field $Y$ does not converge to $\Delta_Y = 2$. We delegate a careful study of these bounds to future work.

We also work out the required tensor structures (under global symmetry) for generic groups $G^n \rtimes S_n$ with $G$ arbitrary. There are at least two motivations for doing so. First, there exist experimentally relevant cases among such models. For example, beyond the cubic ($\mathbb{Z}_2^3 \rtimes S_3$), $MN_{2,2}$ ($O(2)^2 \rtimes S_2$) and $MN_{2,3}$ ($O(2)^3 \rtimes S_3$) models mentioned earlier, there are also the so called tetragonal theories ($D_4^n \rtimes S_n$, with $D_4$ the dihedral group of 8 elements). An incomplete list of references where the reader may find a number of physically motivated examples is [27–31]. A second, broader, motivation stems from our pursuit to better understand the space of three-dimensional CFTs. We hope that our results will enable a variety of future bootstrap studies.

This paper is organized as follows. In the next section we present a quick review of M N models as understood from a Lagrangian point of view and describe our results for global symmetries of the type $G^n \rtimes S_n$. In section 3 we present our numerical bootstrap results. We conclude in section 4. Four technical appendices include calculations of crossing equations in a variety of cases relevant for this work.

## 2 MN and $G^n \rtimes S_n$ symmetries

### 2.1 Definition and Lagrangian description in $d = 4 - \varepsilon$

The global symmetry group $MN_{m,n} = O(m)^n \rtimes S_n$ consists of $n$ copies of the $O(m)$ model that can be permuted with each other. Consider a scalar field multiplet $\phi_i^a$, where the upper index labels the copy and the lower index is an $O(m)$ index. With this field we may write the multi-scalar $MN_{m,n}$-invariant Lagrangian

$$\mathcal{L} = \partial_\mu \phi_i^a \, \partial^\mu \phi_i^a + m^2 \, \phi_i^a \phi_i^a + u(\phi_i^a \phi_i^a)^2 + v \, \delta_{abcd} \, \phi_i^a \phi_i^b \phi_j^c \phi_j^d \,, \tag{1}$$

where summation over repeated indices is implicit and we keep up to quartic terms in $\phi$. The field $\phi_i^a$ transforms in the "defining" representation of $MN_{m,n}$. In other words, the index $a$ has its values permuted by $S_n$, and $O(m)$ acts as $\phi_i^a \to R_{ij} \phi_j^a$ with $R$ including proper and improper rotations. If one were to set $u = 0$ in (1), then (1) would be the Lagrangian of $n$ decoupled $O(m)$ models. Thus, when both $u$ and $v$ are non-zero we obtain a theory describing $n$ coupled $O(m)$ models. Single-coupling deformations of the $O(N)$ theory lead to a total of four fixed points [15, 23, 32], which correspond to the free theory, $n$ coupled $O(m)$ models (i.e. the MN theory), $n$ decoupled $O(m)$ models, and the $O(mn)$ theory.

The generalization of (1) to the $G^n \rtimes S_n$-symmetric case for $G$ arbitrary is straightforward:

$$\mathcal{L} = \partial_\mu \phi_i^a \, \partial^\mu \phi_i^a + m^2 \, \phi_i^a \phi_i^a + u(\phi_i^a \phi_i^a)^2 + \sum_r v_r \, \delta_{abcd} \, T_{ijkl}^r \, \phi_i^a \phi_j^b \phi_k^c \phi_l^d \,, \tag{2}$$

where $r$ labels the four-index fully symmetric invariant tensors $T_{ijkl}^r$ of $G$, to each of which we associate a coupling $v_r$. Only fields from the same copy of $G$ interact when $u = 0$. In tetragonal theories (see e.g. [15]), where $G = D_4$ with $D_4$ the dihedral group of eight elements, we have $r = 1, 2$.

A minor comment is that instead of viewing (1) and (2) as $n$ decoupled $G$-symmetric theories which are consequently coupled via the addition of the $u$ term, we may also view them as deformations of an $O(mn)$ symmetric theory with a term that breaks the symmetry down

to $G^n \rtimes S_n$. Both approaches are fruitful. For example, the authors of [33] study hypercubic theories, where $G = \mathbb{Z}_2$, by performing conformal perturbation theory around the CFT of $n$ decoupled Ising models, whereas the authors of [6] discuss the cubic theory, where $G = \mathbb{Z}_2$ and $n = 3$, by writing it as a deformation of $O(3)$ and then using conformal perturbation theory. Both approaches led to different and interesting results.

## 2.2 The $\phi_i^a \times \phi_j^b$ OPE

The product of two fields transforming in the defining representation of $G^n \rtimes S_n$ is schematically expressed as

$$\phi_i^a \times \phi_j^b \sim \delta^{ab}\delta_{ij}S + \delta_{ij}X^{ab} + I_{1\,ij}^{ab} + \cdots + I_{\mathcal{N}\,ij}^{ab} + Z_{ij}^{ab} + B_{ij}^{ab}\,. \tag{3}$$

Here the lower indices take on values $i, j = 1, \ldots, m$ with $m$ the dimension of the defining irrep of $G$ and the upper indices take on values $a, b = 1, \ldots, n$.[5] The representations on the right-hand side of (3) are to be understood as follows. The last two representations do not exist if $a = b$, whereas all other representations exist only if $a = b$. The first representation, $S$, is the singlet. The representation $X$ is a singlet of $G^n$ and traceless in $S_n$, or to rephrase, it is the fundamental representation of $S_n$. The representations $Z$ and $B$ satisfy $Z_{ij}^{ab} = Z_{ji}^{ba}$ and $B_{ij}^{ab} = -B_{ji}^{ba}$. The remaining irreps $I_i, i = 1, \ldots, \mathcal{N}$, correspond to non-singlet representations of $G$. The dimensions of the irreps $(S, X, I_1, \ldots, I_{\mathcal{N}}, Z, B)$ are $(1, n-1, n\dim R_1, \ldots, n\dim R_{\mathcal{N}}, m^2 \frac{n(n-1)}{2}, m^2 \frac{n(n-1)}{2})$, where $R_1, \ldots, R_{\mathcal{N}}$ are the non-singlet irreps of $G$ that appear in the $\phi_i \times \phi_j$ OPE of $G$. An easy mnemonic rule for obtaining the decomposition in (3) is to decompose onto irreps of $G$ when $a = b$ and to simply symmetrize and antisymmetrize the indices when $a \neq b$.

We may decompose a four point function $\langle \phi_i^a \phi_j^b \phi_k^c \phi_l^d \rangle$ into the irreps that appear in (3). Doing this, we find tensor structures of the global symmetry which in turn determine the sum rules to be eventually studied numerically. Notably, these tensor structures are projectors.[6] The explicit expressions are

$$(P^S)_{ijkl}^{abcd} = \frac{1}{mn}\delta^{ab}\delta^{cd}\delta_{ij}\delta_{kl}\,,$$

$$(P^X)_{ijkl}^{abcd} = \frac{1}{m}\delta_{ij}\delta_{kl}\left(\delta^{abcd} - \frac{1}{n}\delta^{ab}\delta^{cd}\right),$$

$$(P^{I_1})_{ijkl}^{abcd} = \delta^{abcd} P^{R_1}{}_{ijkl}\,,$$

$$\cdots$$

$$(P^{I_{\mathcal{N}}})_{ijkl}^{abcd} = \delta^{abcd} P^{R_{\mathcal{N}}}{}_{ijkl}\,,$$

$$(P^Z)_{ijkl}^{abcd} = \tfrac{1}{2}((\delta^{ac}\delta^{bd} - \delta^{abcd})\delta_{ik}\delta_{jl} + (\delta^{ad}\delta^{bc} - \delta^{abcd})\delta_{il}\delta_{jk})\,,$$

$$(P^B)_{ijkl}^{abcd} = \tfrac{1}{2}((\delta^{ac}\delta^{bd} - \delta^{abcd})\delta_{ik}\delta_{jl} - (\delta^{ad}\delta^{bc} - \delta^{abcd})\delta_{il}\delta_{jk})\,, \tag{4}$$

where the tensors $P^{R_1}{}_{ijkl}, \ldots, P^{R_{\mathcal{N}}}{}_{ijkl}$ correspond to the projectors of the irreps $R_1, \ldots, R_{\mathcal{N}}$ of $G$. Note that all invariant tensors are expressed in terms of (generalized) Kronecker deltas, up to the form of the $P^{R_1}{}_{ijkl}, \ldots, P^{R_{\mathcal{N}}}{}_{ijkl}$ invariant tensors.

We now give a set of explicit examples to clear up any confusion.

### 2.2.1 Example 1: hypercubic theories

We start with hypercubic theories, which have been bootstrapped in [11–14]. In this example, $G = \mathbb{Z}_2$ and so the lower indices $(i, j, k, l)$ may be dropped. Also, in the OPE between

---

[5]For every value $a_*$ of the upper index $a$, $\phi_i^{a_*}$ furnishes the defining irrep of the corresponding $G$ in $G^n$.

[6]Assuming they are normalized properly, which we won't always do. Nevertheless, by slight abuse of terminology we will still call them projectors.

two operators charged under $\mathbb{Z}_2$ only the singlet representation appears, thus we have no "$I$" representations. Hence, the projectors are

$$(P^S)^{abcd} = \frac{1}{n}\delta^{ab}\delta^{cd},$$

$$(P^X)^{abcd} = \delta^{abcd} - \frac{1}{n}\delta^{ab}\delta^{cd},$$

$$(P^Z)^{abcd} = \tfrac{1}{2}((\delta^{ac}\delta^{bd} - \delta^{abcd}) + (\delta^{ad}\delta^{bc} - \delta^{abcd})),$$

$$(P^B)^{abcd} = \tfrac{1}{2}((\delta^{ac}\delta^{bd} - \delta^{abcd}) - (\delta^{ad}\delta^{bc} - \delta^{abcd})),$$

(5)

where the irreps $(S, X, Z, B)$ have dimensions $(1, n-1, \frac{n(n-1)}{2}, \frac{n(n-1)}{2})$, in agreement with the dimensions stated above when $m = 1$.

### 2.2.2 Example 2: MN theories

Next, we consider the MN theories studied in [24] and [25]. In this example we have $G = O(m)$. Since the OPE between two vectors of $O(m)$ exchanges, beyond the singlet, the two-index traceless symmetric $T$ and antisymmetric $A$ irreps, we have $R_1 = T$ and $R_2 = A$. For these irreps we know that $P^{R_1}{}_{ijkl} = \frac{1}{2}(\delta_{ik}\delta_{jl} + \delta_{il}\delta_{jk} - \frac{2}{m}\delta_{ij}\delta_{kl})$ and $P^{R_2}{}_{ijkl} = \frac{1}{2}(\delta_{ik}\delta_{jl} - \delta_{il}\delta_{jk})$. We thus obtain

$$(P^S)^{abcd}_{ijkl} = \frac{1}{mn}\delta^{ab}\delta^{cd}\delta_{ij}\delta_{kl},$$

$$(P^X)^{abcd}_{ijkl} = \frac{1}{m}\delta_{ij}\delta_{kl}\left(\delta^{abcd} - \frac{1}{n}\delta^{ab}\delta^{cd}\right),$$

$$(P^{I_1})^{abcd}_{ijkl} = \delta^{abcd}P^{R_1}{}_{ijkl},$$

$$(P^{I_2})^{abcd}_{ijkl} = \delta^{abcd}P^{R_2}{}_{ijkl},$$

$$(P^Z)^{abcd}_{ijkl} = \tfrac{1}{2}((\delta^{ac}\delta^{bd} - \delta^{abcd})\delta_{ik}\delta_{jl} + (\delta^{ad}\delta^{bc} - \delta^{abcd})\delta_{il}\delta_{jk}),$$

$$(P^B)^{abcd}_{ijkl} = \tfrac{1}{2}((\delta^{ac}\delta^{bd} - \delta^{abcd})\delta_{ik}\delta_{jl} - (\delta^{ad}\delta^{bc} - \delta^{abcd})\delta_{il}\delta_{jk}),$$

(6)

where the representations $(S, X, I_1, I_2, Z, B)$ have dimensions

$$\left(1, \; n-1, \; n\frac{(m-1)(m+2)}{2}, \; n\frac{m(m-1)}{2}, m^2\frac{n(n-1)}{2}, \; m^2\frac{n(n-1)}{2}\right),$$

(7)

in agreement with the general formulas given earlier and the computations of [24]. In what follows we will rename $I_1$ and $I_2$ as $Y$ and $A$ to keep the same names as [24].

### 2.2.3 Example 3: Tetragonal theories and their generalizations

We may also study a generalization of the tetragonal theories bootstrapped in [24]. In this example we have $G = \mathbb{Z}_2^m \rtimes S_m$. Thus, similarly,

$$(P^S)^{abcd}_{ijkl} = \frac{1}{mn}\delta^{ab}\delta^{cd}\delta_{ij}\delta_{kl},$$

$$(P^X)^{abcd}_{ijkl} = \frac{1}{m}\delta_{ij}\delta_{kl}\left(\delta^{abcd} - \frac{1}{n}\delta^{ab}\delta^{cd}\right),$$

$$(P^{I_1})^{abcd}_{ijkl} = \delta^{abcd}\left(\delta_{ijkl} - \frac{1}{m}\delta_{ij}\delta_{kl}\right),$$

$$(P^{I_2})^{abcd}_{ijkl} = \delta^{abcd}\tfrac{1}{2}(\delta_{ik}\delta_{jl} + \delta_{il}\delta_{jk} - 2\delta_{ijkl}),$$

$$(P^{I_3})^{abcd}_{ijkl} = \delta^{abcd}\tfrac{1}{2}(\delta_{ik}\delta_{jl} - \delta_{il}\delta_{jk}),$$

$$(P^Z)^{abcd}_{ijkl} = \tfrac{1}{2}((\delta^{ac}\delta^{bd} - \delta^{abcd})\delta_{ik}\delta_{jl} + (\delta^{ad}\delta^{bc} - \delta^{abcd})\delta_{il}\delta_{jk}),$$

$$(P^B)^{abcd}_{ijkl} = \tfrac{1}{2}((\delta^{ac}\delta^{bd} - \delta^{abcd})\delta_{ik}\delta_{jl} - (\delta^{ad}\delta^{bc} - \delta^{abcd})\delta_{il}\delta_{jk}),$$

(8)

where now the irreps $(S, X, I_1, I_2, I_3, Z, B)$ have dimensions

$$\left(1,\ n-1,\ n(m-1),\ n\frac{m(m-1)}{2},\ n\frac{m(m-1)}{2}, m^2\frac{n(n-1)}{2},\ m^2\frac{n(n-1)}{2}\right). \tag{9}$$

These indeed agree with and generalize to arbitrary $m$ the results of [24].

## 2.3 The $\phi_i^a \times X^{bc}$ OPE

In our mixed correlator system we will need to consider the OPE of $\phi$ with $X$. A first observation is that the decomposition of this OPE does not depend on $G$ because $X$ is a singlet of $G$. We also know that $\phi \times X$ should exchange $\phi$, as dictated by self-consistency of the three-point function, i.e. calculating the three point function with different OPEs should give the same result. Lastly, the existence of the defining representation ($\phi$) on the right hand side of the OPE implies the existence also of an antisymmetric representation ($\bar{A}$) on the right-hand side, see e.g. [34, Chapter 4.3]. If one now counts the dimensions of these two irreps ($\phi$ and $\bar{A}$), their sum turns out equal to the product of the dimensions on the left-hand side of the OPE. Thus, the OPE is fully decomposed. Schematically, the decomposition looks like

$$\phi_i^a X^{bc} \sim (P^\phi)^{abcdef} \phi_i^d X^{ef} + (P^{\bar{A}})^{abcdef} \phi_i^d X^{ef}, \tag{10}$$

where

$$(P^\phi)^{abcdef} = \delta_{abcdef} - \frac{1}{n}(\delta^{bc}\delta^{adef} + \delta^{ef}\delta^{abcd}) + \frac{1}{n^2}\delta^{ad}\delta^{bc}\delta^{ef} \tag{11}$$

and

$$(P^{\bar{A}})^{abcdef} = -\delta^{abcdef} + \frac{n-1}{n}\delta^{ad}\delta^{bcef} + \frac{1}{n}(\delta^{bc}\delta^{adef} + \delta^{ef}\delta^{abcd}) - \frac{1}{n}\delta^{ad}\delta^{bc}\delta^{ef}. \tag{12}$$

It is a straightforward exercise to contract the tensors quoted above in (10) and obtain the expected representation dimensions. Note that (11) and (12) are the projectors we need in order to obtain the sum rules from the corresponding crossing equations. An additional check one can perform is test the above decomposition for some discrete group $G$ in which the decompositions are known, and then remembering that the formula is $G$-independent one has proved the result.

## 2.4 The $X^{ab} \times X^{cd}$ OPE

It is useful to remember at this point that the $X$ representation is simply the fundamental representation of $S_n$, and so its OPE decomposition is already known from e.g. [35]; see also the later works [11] and [12]. Hence, both projectors and crossing equation sum rules are known. Note that the fundamental of $S_n$ is usually written with one index. For the explicit map between the one- and two- index notations see [34].

## 2.5 The $n \geqslant 4\ Z_{ij}^{ab} \times Z_{kl}^{cd}$ OPE

In this work we will also consider the $\langle ZZZZ \rangle$ single correlator bootstrap for values of $n \geqslant 4$ (and $m \geqslant 2$). This is because we can then probe various large parameter limits, such as the $m \to \infty$ and $n \to \infty$ limits.[7] Probing such limits may give hints about the existence or not of possible perturbative expansions of theories satisfying these single correlator constraints. One such example could be a theory where the order parameter field transforms in the $Z$ irrep. A similar study was performed for traceless tensors of $O(n)$ in [26] and adjoints of $SU(n)$ in [36],

---

[7]For $m \geqslant 2$ and $n \geqslant 4$ the sum rules take their most general form, valid for any such $m$ and $n$. In Appendix B we additionally work out the $n = 2$ case, which is special.

where theories with order parameter fields in these representations were analysed. The reader may skip this subsection on a first reading since it is rather technical, and return on a second read-through.

For $n \geqslant 4$ one finds 21 irreducible representations on the right-hand side of the $n \geqslant 4$ $Z_{ij}^{ab} \times Z_{kl}^{cd}$ OPE.[8] We present the projectors of the 21 irreps in Appendix C, where we also explain the generalization to arbitrary group $G$. When all indices $a$, $b$, $c$ and $d$ are different, we have three exchanged irreps. It is convenient to define $T_{ij,kl}^{ab,cd} = Z_{ij}^{ab} Z_{kl}^{cd} + Z_{kl}^{cd} Z_{ij}^{ab}$ and $\bar{T}_{ij,kl}^{ab,cd} = Z_{ij}^{ab} Z_{kl}^{cd} - Z_{kl}^{cd} Z_{ij}^{ab}$, which obviously satisfy $T_{ij,kl}^{ab,cd} = T_{kl,ij}^{cd,ab}$ and $\bar{T}_{ij,kl}^{ab,cd} = -\bar{T}_{kl,ij}^{cd,ab}$. With this in mind, we have

$$Z_{ij}^{ab} \times Z_{kl}^{cd} \sim T_{ij,kl}^{ab,cd} + \bar{T}_{ij,kl}^{ab,cd} . \tag{13}$$

The right-hand side may now be further decomposed by symmetrizing and antisymmetrizing indices,

$$Z_{ij}^{ab} \times Z_{kl}^{cd} \sim (T_{ij,kl}^{ab,cd} + T_{ik,jl}^{ac,bd} + T_{il,jk}^{ad,bc}) + (T_{ij,kl}^{ab,cd} - T_{ik,jl}^{ac,bd}) + (T_{ij,kl}^{ab,cd} - T_{il,jk}^{ad,bc}) \tag{14}$$
$$+ (\bar{T}_{ij,kl}^{ab,cd} + \bar{T}_{ik,jl}^{ac,bd}) + (\bar{T}_{ij,kl}^{ab,cd} + \bar{T}_{il,jk}^{ad,bc}) + (\bar{T}_{ij,kl}^{ab,cd} - \bar{T}_{ik,jl}^{ac,bd}) + (\bar{T}_{ij,kl}^{ab,cd} - \bar{T}_{il,jk}^{ad,bc}),$$

which may also be rewritten as

$$Z_{ij}^{ab} \times Z_{kl}^{cd} \sim TotS_{ijkl}^{abcd} + BB_{il,jk}^{ad,bc} + BB_{ik,jl}^{ac,bd} \tag{15}$$
$$+ BZ_{il,jk}^{ad,bc} + BZ_{ik,jl}^{ac,bd} + ZB_{il,jk}^{ad,bc} + ZB_{ik,jl}^{ac,bd} ,$$

where $TotS$ is totally symmetric in all pairs of indices,[9] whereas for $BB$ and $BZ$ we have the relations $BB_{ik,jl}^{ac,bd} = -BB_{ki,jl}^{ca,bd} = -BB_{ik,lj}^{ac,db} = BB_{jl,ik}^{bd,ac}$ and $BZ_{ik,jl}^{ac,bd} = -BZ_{ki,jl}^{ca,bd} = +BZ_{ik,lj}^{ac,db} = -BZ_{jl,ik}^{bd,ac}$. Lastly, notice that $BZ$ and $ZB$ are the same representation since $BZ_{ij,kl}^{ab,cd} = ZB_{kl,ij}^{cd,ab}$.

Next we must consider the representations where one or more of the copy (i.e. upper) indices are equal in $Z_{ij}^{ab} \times Z_{kl}^{cd}$. These are rather simple: if a pair of copy indices are equal we decompose onto irreps of that copy of $G$, whereas if two copy indices are different we simply symmetrize and antisymmetrize. Let us give some examples. Consider $a = c$ and $b \neq d$. We have

$$Z_{ij}^{ab} \times Z_{kl}^{cd} \sim (Z_{ij}^{ab} Z_{kl}^{cd} + Z_{il}^{ad} Z_{kj}^{cb}) + (Z_{ij}^{ab} Z_{kl}^{cd} - Z_{il}^{ad} Z_{kj}^{cb}) \sim RZ_{ik,jl}^{ac,bd} + RB_{ik,jl}^{ac,bd} , \tag{16}$$

where the part $R$ simply signifies that $a = c$ but is not an irrep. To decompose $R$ onto irreps we must decompose the indices $i$ and $k$ onto $G$ irreps and then subtract the trace of copies from the singlet of $G$:

$$RZ_{ik,jl}^{ac,bd} \sim \left(RZ_{ik,jl}^{ac,bd} + RZ_{ki,jl}^{ac,bd} - \frac{2}{m}\delta_{ik}RZ_{mm,jl}^{ac,bd}\right) + (RZ_{ik,jl}^{ac,bd} - RZ_{ki,jl}^{ac,bd}) + \frac{2}{m}\delta_{ik}RZ_{mm,jl}^{ac,bd}, \tag{17}$$

and

$$\delta_{ik}RZ_{mm,jl}^{ac,bd} \sim \delta_{ik}\left(RZ_{mm,jl}^{ac,bd} - \frac{1}{n-2}\delta_{ac}RZ_{mm,jl}^{ee,bd}\right) + \frac{1}{n-2}\delta_{ik}\delta_{ac}RZ_{mm,jl}^{ee,bd} \sim XZ_{ik,jl}^{ac,bd} + SZ_{ik,jl}^{ac,bd} , \tag{18}$$

where we have absorbed factors of Kronecker deltas in the representations in order to simplify the notation. Combining all this information together we find

$$RZ_{ik,jl}^{ac,bd} \sim YZ_{ik,jl}^{ac,bd} + AZ_{ik,jl}^{ac,bd} + XZ_{ik,jl}^{ac,bd} + SZ_{ik,jl}^{ac,bd} , \tag{19}$$

---

[8]Note that for $n = 3$ there are fewer irreps. Also, the invariant tensor $\delta_{abcdefgh}$ becomes expressible in terms of tensors with fewer indices, see e.g. [13]. We omit an analysis of the $n = 3$ case in the present work since we are interested in the large parameter limits.

[9]Remember that permuting $a$ with $b$ means we must permute $i$ with $j$, hence we do not permute indices, but pairs of them.

which we remind the reader is for $a = c$ and $b \neq d$. For the full expression one needs to consider all possible combinations.

The last case we must consider is when we have pairs of copy indices equal, e.g. $a = c$ and $b = d$. We must again decompose onto irreps of $G$. Let us start by first decomposing $j$ and $l$ onto $G$ irreps:

$$Z_{ij}^{ab} \times Z_{kl}^{cd} \sim RR_{ik,jl}^{ac,bd} \sim \left( RR_{ik,jl}^{ac,bd} + RR_{ik,lj}^{ac,bd} - \frac{2}{m} \delta_{jl} RR_{ik,mm}^{ac,bd} \right) + \left( RR_{ik,jl}^{ac,bd} - RR_{ik,lj}^{ac,bd} \right) + \frac{2}{m} \delta_{jl} RR_{ik,mm}^{ac,bd} . \tag{20}$$

This is rewritten as

$$Z_{ij}^{ab} \times Z_{kl}^{cd} \sim RY_{ik,jl}^{ac,bd} + RA_{ik,jl}^{ac,bd} + \frac{2}{m} \delta_{jl} RR_{ik,mm}^{ac,bd} , \tag{21}$$

where we use $RR_{ik,jl}^{ac,bd}$ to denote an object that is not an irrep but has $a = c$ and $b = d$. We will now focus on the last term, which is the most complicated. We decompose the $i$ and $k$ indices onto $G$ irreps,

$$RR_{ik,mm}^{ac,bd} \sim \left( RR_{ik,mm}^{ac,bd} + RR_{ki,mm}^{ac,bd} - \frac{2}{m} \delta_{ik} RR_{nn,mm}^{ac,bd} \right) + \left( RR_{ik,mm}^{ac,bd} - RR_{ki,mm}^{ac,bd} \right) + \frac{2}{m} \delta_{ik} RR_{nn,mm}^{ac,bd} , \tag{22}$$

which is equivalent to

$$RR_{ik,mm}^{ac,bd} \sim YR_{ik,mm}^{ac,bd} + AR_{ik,mm}^{ac,bd} + \frac{2}{m} \delta_{ik} RR_{nn,mm}^{ac,bd} . \tag{23}$$

The last term is again the most complicated.[10] Defining $T^{ac,bd} = RR_{nn,mm}^{ac,bd}$, we subtract all possible traces to construct irreducible objects:

$$\begin{aligned} T^{ac,bd} &\sim T^{ac,bd} - \frac{1}{n-2}(\delta_{bd} T^{ac,ee} + \delta_{ac} T^{ee,bd}) + \frac{\delta_{ac} \delta_{bd}}{(n-1)(n-2)} T^{ee,ff} \\ &+ \frac{\delta_{bd}}{n-2}\left( T^{ac,ee} - \frac{\delta_{ac}}{n} T^{ee,ff} \right) + \frac{\delta_{ac}}{n-2}\left( T^{ee,bd} - \frac{\delta_{bd}}{n} T^{ee,ff} \right) \\ &- \delta_{ac} \delta_{bd}\left( \frac{1}{(n-1)(n-2)} - \frac{2}{n(n-2)} \right) T^{ee,ff} \\ &\sim \bar{X}^{ac,bd} + XS^{ac,bd} + SX^{ac,bd} + \delta_{ac} \delta_{bd} S , \end{aligned} \tag{24}$$

where again we have absorbed various factors in the definitions of the irreps. The other irreps follow rather straightforwardly, hence we omit a detailed analysis. In conclusion we find the 21 irreps $(S, XS, \bar{X}, XY, XA, SY, SA, YY, AA, YA, BB, TotS, XB, XZ, SB, SZ, YB, YZ, AB, AZ, BZ)$ with respective dimensions $(S : 1, XS : n-1, \bar{X} : n(n-3)/2, XY : n(n-2)(m-1)(m+2)/2, XA : n(n-2)(m-1)m/2, SY : n(m-1)(m+2)/2, SA : n(m-1)m/2, YY : n(n-1)(m+2)^2(m-1)^2/8, AA : n(n-1)(m-1)^2 m^2/8, YA : n(n-1)(m-1)^2 m(m+2)/4, BB : 2m^4 n(n-1)(n-2)(n-3)/(4!), TotS : m^4 n(n-1)(n-2)(n-3)/(4!), XB : (n-3)m^2 n(n-1)/2, XZ : (n-3)m^2 n(n-1)/2, SB : m^2 n(n-1)/2, SZ : m^2 n(n-1)/2, YB : (m+2)(m-1)m^2 n^2(n-1)/4, YZ : (m+2)(m-1)m^2 n^2(n-1)/4, AB : (m-1)m^3 n^2(n-1)/4, AZ : (m-1)m^3 n^2(n-1)/4, BZ : 3m^4 n(n-1)(n-2)(n-3)/(4!))$.

---

[10]Since, for example, we know $YR_{ik,mm}^{ac,bd} \sim YR_{ik,mm}^{ac,bd} - \delta^{bd} YR_{ik,mm}^{ac,ee}/(n-1) + \delta^{bd} YR_{ik,mm}^{ac,ee}/(n-1)$ or stated in terms of the irreps $YR_{ik,mm}^{ac,bd} \sim YX_{ik,jl}^{ac,bd} + YS_{ik,jl}^{ac,bd}$ where again we have absorbed Kronecker deltas and numerical factors in the definitions of the irreps for simplicity.

# 3 Numerical results

With the group theory outlined above and the sum rules given in the appendices we are ready to start obtaining numerical results. We start by probing the system of four point correlators obtained by considering all combinations of external operators $\phi_i^a$ and $X^{bc}$. We then conclude the chapter by presenting bounds pertaining to four point functions of symmetric tensors $Z_{ij}^{ab}$.

## 3.1 Islands close to the unitarity bound

In this section we study the portion of parameter space "close" to the unitarity bound. By this we refer to values of the order parameter scaling dimensions roughly around 0.5-0.53; these correspond to typical values found in fixed points of multi-scalar theories. However, this does not necessarily mean that the theories we find are due to the fixed point of some multi-scalar Lagrangian description. For $n$ sufficiently large we find kinks, and corresponding islands, that converge to the expected values, that is $\Delta_\phi = \Delta_\phi^{O(m)}$ and $\Delta_X = \Delta_S^{O(m)}$.[11] Here $\Delta_\phi^{O(m)}$ is the scaling dimension of the order parameter field and $\Delta_S^{O(m)}$ is the scaling dimension of the lowest lying scalar singlet, both in the theory of $n$ decoupled $O(m)$ models. For smaller values of $n$, which also correspond to the phenomenologically interesting values, it is not clear whether the kinks (and their islands) correspond to the ordinary $d = 4 - \varepsilon$ fixed points or are some new hitherto unknown CFTs.

We separate our results into two groups depending on how we obtain the islands. In the case where the islands are found in the $\Delta_\phi$-$\Delta_X$ plane of parameter space, we impose gaps on certain sectors mainly guided by the extremal functional method [37, 38].[12] Since these gaps are motivated by the extremal functional, their choice is not rigorous. Nevertheless, we believe that even in absence of rigor our islands are sufficient to motivate that something particularly interesting takes place in the corresponding region of parameter space. We also present islands in the $\Delta_\phi$-$\Delta_S$ plane, specifically in the phenomenologically interesting cases $O(2)^2 \rtimes S_2$ and $O(2)^3 \rtimes S_3$. For these cases, in addition to assumptions described above, we demand the saturation of certain exclusion bounds. More specifically, we demand that the exclusion bounds in the $X$ sector[13] are saturated.

Obtaining islands by demanding saturation of exclusion bounds (in our case the $X$ bound) is somewhat morally similar to the extremal functional method where one may again demand saturation of a bound to then find an approximate spectrum that corresponds to it. However, we believe that our method minimizes the probability that a zero of the extremal function is spurious. This is because in our approach we do not try to find approximate zeroes of the functional, but instead show that the positions of these zeroes cannot be excluded by the bootstrap algorithm (whereas the parameter space surrounding them can be). Also, in conjunction with additional assumptions we can provide a minimum and maximum value for $\Delta_\phi$. Lastly, we note that we have found these islands to depend smoothly on the precise position of the $X$ bound. In other words, if $\Delta_X$ were to change a little there would be no major changes in the corresponding $\Delta_\phi$-$\Delta_S$ plane island. We confirmed this behavior while working on [14], albeit for a slightly different symmetry, namely $\mathbb{Z}_2^n \rtimes S_n$. Another way to

---

[11]Whereas for the scalar singlet one has and $\Delta_S = d - \Delta_S^{O(m)}$, although we do not present results for this in the present work.

[12]One can also try to justify the gaps from the large $n$ point of view. For example, we know that the leading $X^{ab}$ operator has scaling dimension $\Delta_X = \Delta_S^{O(m)} + O(1/n)$, so we expect the subleading operator to have dimension either $\Delta_{X'} = \Delta_{S'}^{O(m)} + O(1/n)$ or $\Delta_{X'} = 2\Delta_S^{O(m)} + O(1/n)$ or $\Delta_{X'} = \Delta_S^{O(m)} + 2 + O(1/n)$ (remember $X^{ab} \sim (\delta^{abcd} - \frac{1}{n}\delta^{ab}\delta^{cd})\phi_i^c\phi_i^d$ in the weakly coupled limit). All these satisfy $\Delta_{X'} > 3$ at $n \to \infty$. The main issue with this line of reasoning though is that we do not have the explicit corrections to subleading order in $1/n$.

[13]These are the exclusion plots that display kinks/changes of slope.

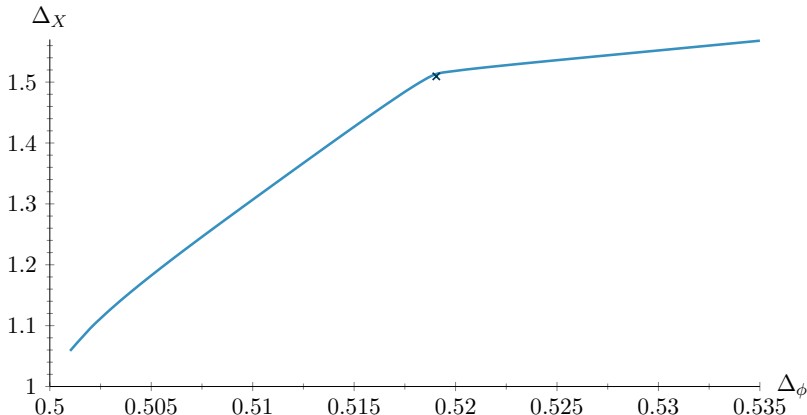

Figure 1: Single correlator $MN_{2,100}$ exclusion bound. The cross is the $O(2)$ model [5]. The line corresponds to the maximum allowed scaling dimension of the first scalar $X$ operator as a function of the scaling dimension of the order parameter operator. In qboot we used $\Lambda = 45$, $\ell = \{0,\ldots,50,55,56,59,60,64,65,69,70,74,75,79,80,84,85,89,90\}$ and $\nu_{\max} = 20$.

think of our method for obtaining islands in the $\Delta_\phi$-$\Delta_S$ plane, is that we apply the usual bootstrap algorithm, but to a specific slice of parameter space. This slice is precisely the one that maximizes the scaling dimension $\Delta_X$.

We note that for our numerical calculations in the present work we use qboot [39][14] and SDPB [41,42]. Some of the spectrum calculations were performed with PyCFTBoot [43].

### 3.1.1 Islands in the $\Delta_\phi$-$\Delta_X$ plane of parameter space

In Figs. 2 and 4 we plot the islands corresponding respectively to the kinks in Figs. 1 and 3. Note that we do not present an island in the $\Delta_\phi$-$\Delta_X$ plane for $n = 2$, even though it is phenomenologically interesting. This is because up to $\Lambda = 60$ in qboot the allowed region obtained is very large. For Fig. 2 and Fig. 1 we see good agreement with the perturbative expectation.[15] That is, as $n \to \infty$ we expect $\Delta_X \to \Delta_S^{O(m)}$, hence for Fig. 2 in particular we have $\Delta_X \sim \Delta_S^{O(2)} = 1.50946(22)$ and $\Delta_\phi \sim \Delta_\phi^{O(2)} = 0.519050(40)$ [5]. The interested reader is referred to [44] for the large $n$ limit in the case $G = \mathbb{Z}_2$ or to [45] for a more recent reference. For Figs. 4 and 3 the situation is more complicated. We cannot make any conclusive statements with regards to the theory captured by this island.

### 3.1.2 Islands in the $\Delta_\phi$-$\Delta_S$ plane of parameter space

The motivation behind studying the islands in the $\Delta_\phi$-$\Delta_S$ plane of parameter space is twofold. On the one hand, the $\Delta_\phi$-$\Delta_S$ plane of parameter space is the most immediately relevant one in terms of critical exponents since $\nu = 1/(3-\Delta_S)$ and $\beta = \Delta_\phi/(3-\Delta_S)$. On the other, this way (i.e. by demanding saturation of the $X$ sector exclusion bound) we can study specifically the theory that creates the kink by excluding the rest of parameter space. We do again emphasize that this approach is not rigorous in the usual sense of the term as used in the bootstrap.

With these considerations in mind, in Figs. 5 and 9 we obtain islands pertaining to the theories saturating the corresponding $MN_{2,3}$ and $MN_{2,2}$ $X$ sector bounds. Note that in Fig. 6

---

[14]We also encourage the reader to see [40], which automates the derivation of sum rules, although we did not use it for the present paper since our groups of interest were not supported.

[15]SRK thanks Bernardo Zan for pointing out aspects of the large $n$ description.

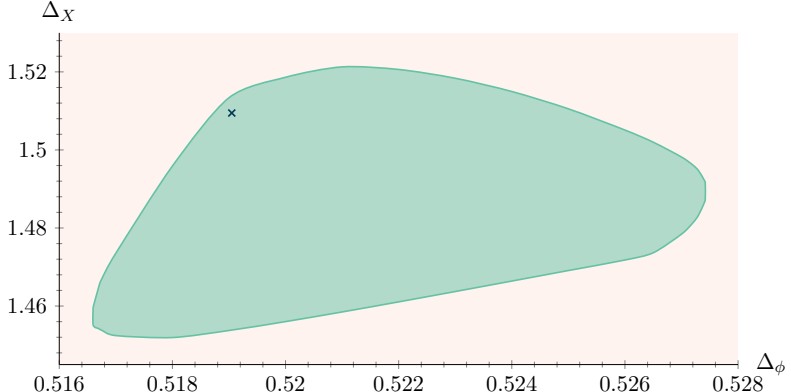

Figure 2: $MN_{2,100}$ island corresponding to the kink in Fig. 1 obtained using the mixed $\phi$-$X$ system of correlators. The cross is the $O(2)$ model [5]. The island assumes that the second scalar $X$ and spin-1 $A$ operators have scaling dimensions that satisfy $\Delta \geqslant 3.0$, and the first spin-2 singlet after the stress tensor has a dimension that satisfies $\Delta \geqslant 4.0$. The first spin-1 $A$ operator satisfies $\Delta_A = 2.0$ since it is the conserved vector of $O(2)$, and the first spin-2 $S$ operator satisfies $\Delta_{T_{\mu\nu}} = 3.0$. We also fixed the ratio of the OPE coefficients with which the stress tensor appears, $\lambda_{\phi\phi T_{\mu\nu}}/\lambda_{XXT_{\mu\nu}} = \Delta_\phi/\Delta_X$. Lastly, we imposed $\Delta_{\phi'} \geqslant 1.0$ for the second operator in the defining representation. In qboot we used $\Lambda = 27$, $\ell = \{0, \ldots, 50, 55, 56, 59, 60, 64, 65, 69, 70, 74, 75, 79, 80, 84, 85, 89, 90\}$ and $\nu_{\max} = 25$.

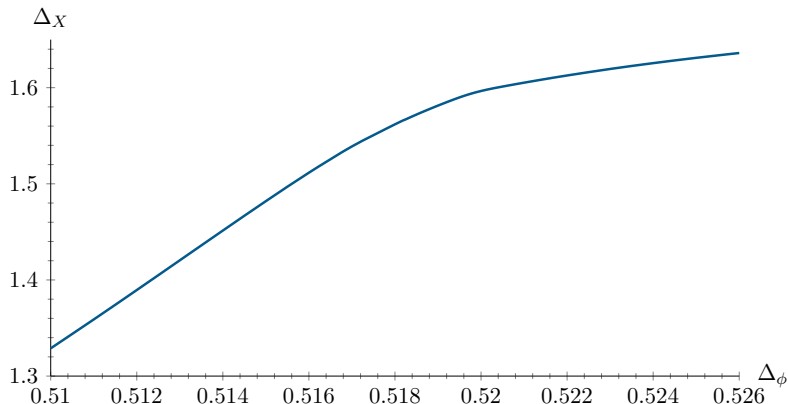

Figure 3: Single-correlator $MN_{2,3}$ exclusion bound. The line corresponds to the maximum allowed scaling dimension of the first scalar $X$ operator as a function of the scaling dimension of the order parameter operator. In qboot we used $\Lambda = 45$, $\ell = \{0, \ldots, 50, 55, 56, 59, 60, 64, 65, 69, 70, 74, 75, 79, 80, 84, 85, 89, 90\}$ and $\nu_{\max} = 20$.

we present again the $MN_{2,3}$ island but with much larger gaps in order to demonstrate that as our gaps approach the values predicted by the extremal functional method the islands become more smooth. Lastly, in Fig. 7 we explicitly show the overlap of these two figures.

Additionally, note that the rightmost tip of the left blue island in Fig. 9 extends to the $\Delta_\phi$ of the kink of Fig. 8. Thus, the putative theory that lives at that kink is allowed under the assumptions mentioned in the caption of Fig. 9. Given that this is a marginal case, however, further numerical work with stronger numerics and more refined methods is required to provide further clarity.

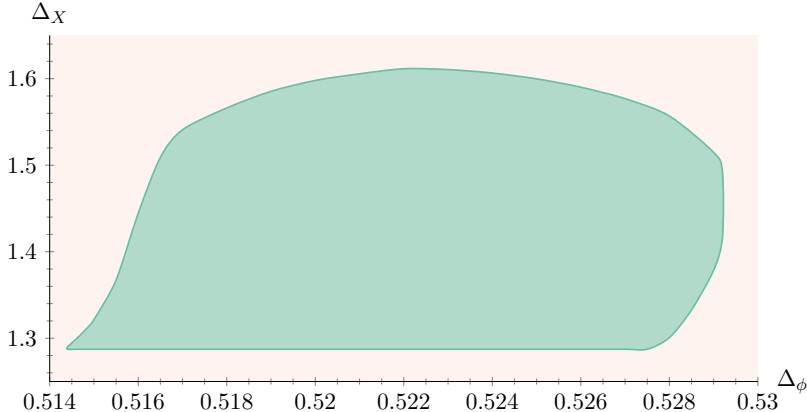

Figure 4: $MN_{2,3}$ island corresponding to the kink in Fig. 3 obtained using the mixed $\phi$-$X$ system of correlators. The island assumes that the second scalar $X$ and spin-1 $A$ operators have scaling dimensions that satisfy $\Delta \geqslant 3.0$ and the first spin-2 singlet after the stress tensor has a dimension that satisfies $\Delta \geqslant 4.0$. The first spin-1 $A$ operator satisfies $\Delta_A = 2.0$ since it is the conserved vector of $O(2)$, and the first spin-2 $S$ operator satisfies $\Delta_{T_{\mu\nu}} = 3.0$. We also fixed the ratio of the OPE coefficients with which the stress tensor appears, $\lambda_{\phi\phi T_{\mu\nu}}/\lambda_{XXT_{\mu\nu}} = \Delta_\phi/\Delta_X$. Lastly, we imposed $\Delta_{\phi'} \geqslant 1.0$ for the next-to-leading operator in the defining representation. To obtain this figure with qboot [39] we used $\Lambda = 27$, $\ell = \{0, \ldots, 50, 55, 56, 59, 60, 64, 65, 69, 70, 74, 75, 79, 80, 84, 85, 89, 90\}$ and $\nu_{\max} = 25$.

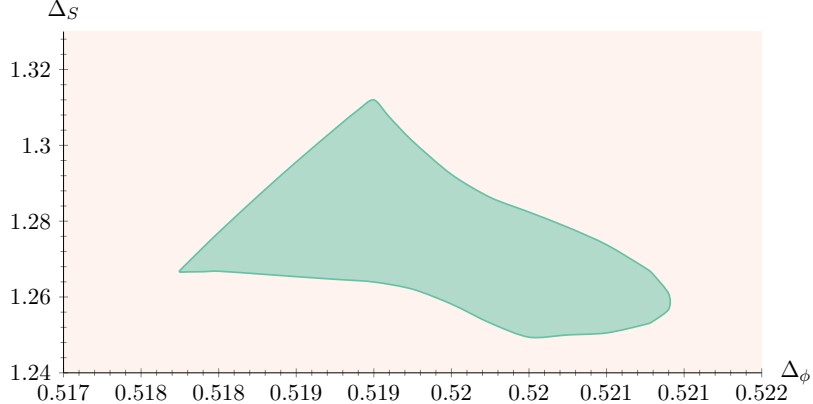

Figure 5: $MN_{2,3}$ island corresponding to the theory saturating Fig. 3 obtained using the mixed $\phi$-$X$ system of correlators. To obtain the island we imposed $\Delta_{S'} \geqslant 3.0$, $\Delta_A = 2.0$ ($O(2)$ conserved vector), $\Delta_{A'} \geqslant 3.0$, $\Delta_{X'} \geqslant 3.0$ and $\Delta_{\phi'} \geqslant 1.0$. Lastly, we assumed that the first scalar $X$ operator saturates the bound of Fig. 3. To obtain this figure with qboot we used $\Lambda = 27$, $\ell = \{0, \ldots, 50, 55, 56, 59, 60, 64, 65, 69, 70, 74, 75, 79, 80, 84, 85, 89, 90\}$ and $\nu_{\max} = 25$.

## 3.2 Islands further away from the unitarity bound

In [24] more kinks further away from the unitarity bound were observed, such as the one in Fig. 10. These were subsequently studied in [25]. In this work we probe them using two different strategies. The first strategy is to bootstrap them directly, i.e. to demand that the exclusion plot with the kink is saturated and impose assumptions inspired by the extremal functional

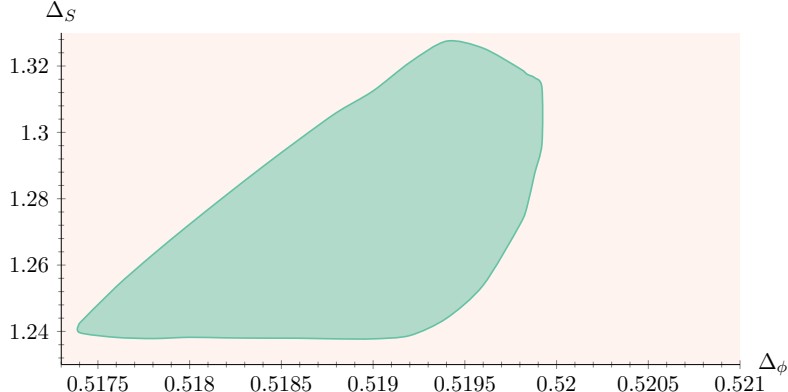

Figure 6: $MN_{2,3}$ island corresponding to the theory saturating Fig. 3 obtained using the mixed $\phi$-$X$ system of correlators. To obtain the island we imposed $\Delta_{S'} \geqslant 3.5$, $\Delta_A = 2.0$ ($O(2)$ conserved vector), $\Delta_{A'} \geqslant 3.9$, $\Delta_{X'} \geqslant 3.0$ and $\Delta_{\phi'} \geqslant 1.5$. Lastly, we assumed that the first $X$ operator saturates the bound of Fig. 3. To obtain this figure with qboot we used $\Lambda = 20$, $\ell = \{0, \ldots, 50, 55, 56, 59, 60, 64, 65, 69, 70, 74, 75, 79, 80, 84, 85, 89, 90\}$ and $\nu_{\max} = 25$.

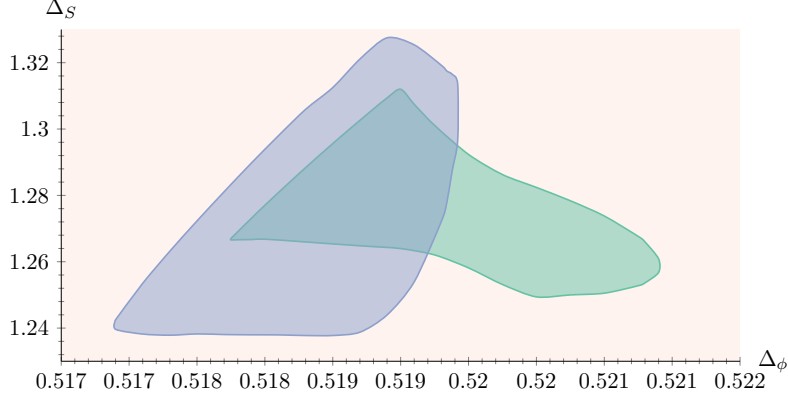

Figure 7: $MN_{2,3}$ islands of the last two figures, namely Fig. 5 and Fig. 6, in a combined plot. We emphasize that the blue island is computed at a lower $\Lambda$, and that is why it appears larger on the left despite the stronger spectrum assumptions used to obtain it compared to the green island. The stronger spectrum assumptions have an effect on the right part of the island.

method to obtain an isolated allowed region. The second strategy is to try and study the kinks indirectly, based on the observation made in [25] that the spectrum at these kinks contained operators in other representations very close to the unitarity bound, namely fields in the $Y$ and $Z$ irreps. Hence, since the numerical bootstrap tends to be stronger closer to the unitarity bound, we can bootstrap these fields instead of the initial $\phi$ field we were studying. We only give a brief example of this strategy in this work, and leave a more complete treatment to future work.

An interesting observation about the second kinks is that, to our knowledge, they have appeared in all works studying scalars involving the group $S_n$. More specifically they have appeared in [11] ($S_n \times \mathbb{Z}_2$), [34] ($\mathbb{Z}_2^n \rtimes S_n$) and [25] ($O(m)^n \rtimes S_n$). Notably, they always appear in what we call the $X$ sector in this work. Second kinks have also appeared in other theories [26, 36, 46–49].

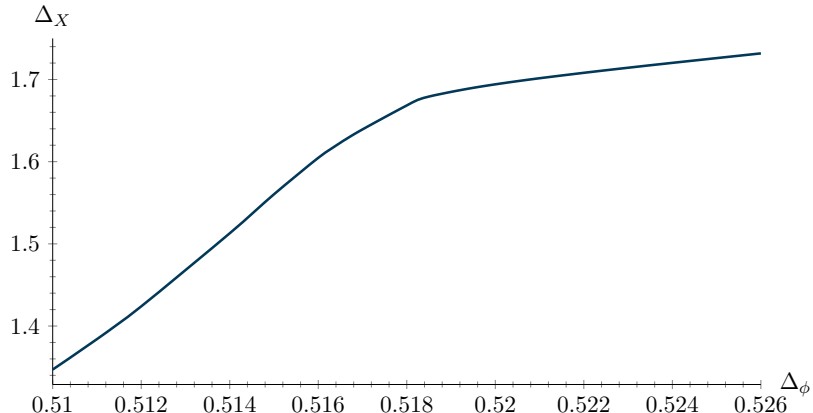

Figure 8: Mixed correlator ($\phi$-$X$) $MN_{2,2}$ exclusion bound. The line corresponds to the maximum allowed scaling dimension of the first scalar $X$ operator as a function of the scaling dimension of the order parameter operator. Here we used the mixed- instead of the single-correlator system, since in this particular case the sum rules are simpler and hence numerically cheaper, than the $n \geqslant 3$ cases. To obtain this figure with qboot we used $\Lambda = 45$, $\ell = \{0, \ldots, 50, 55, 56, 59, 60, 64, 65, 69, 70, 74, 75, 79, 80, 84, 85, 89, 90\}$ and $\nu_{\max} = 20$.

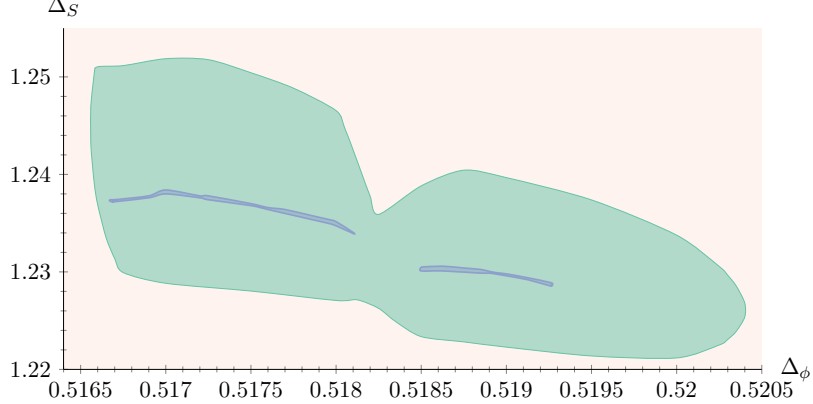

Figure 9: $MN_{2,2}$ island corresponding to the theory saturating Fig. 8 obtained using the mixed $\phi$-$X$ system of correlators. To obtain the island we imposed $\Delta_{S'} \geqslant 3.0$, $\Delta_A = 2.0$ ($O(2)$ conserved vector), $\Delta_{A'} \geqslant 3.0$, $\Delta_{X'} \geqslant 3.0$ and $\Delta_{\phi'} \geqslant 1.0$. Lastly, we assumed that the first scalar $X$ operator saturates the bound of Fig. 8. In qboot we used $\Lambda = 35$, $\ell = \{0, \ldots, 50, 55, 56, 59, 60, 64, 65, 69, 70, 74, 75, 79, 80, 84, 85, 89, 90\}$ and $\nu_{\max} = 25$. We also display (in blue) two narrow islands that correspond to the allowed region that remains if we choose to saturate the $\Delta_X$ bound at $\Lambda = 35$ instead of $\Lambda = 45$ (Fig. 8 is obtained at $\Lambda = 45$).

Note that in Fig. 11 we were still able to obtain an island even if we imposed $\Delta_{S'} \geqslant 5.0$. This is not allowed, though, for unitary CFTs in 3D, as can be seen from [50, Fig. 3].[16] In other words, the bootstrap seems to be completely insensitive to that operator. This is not the first time we observe this phenomenon. For example when working on [51], where we had concrete large $n$ predictions to compare to, the bootstrap seemed to be completely insensitive to the second scalar singlet at the antichiral fixed point for $m = 2$ and $n = 10$, even though all

---

[16]We thank Alessandro Vichi for bringing this to our attention

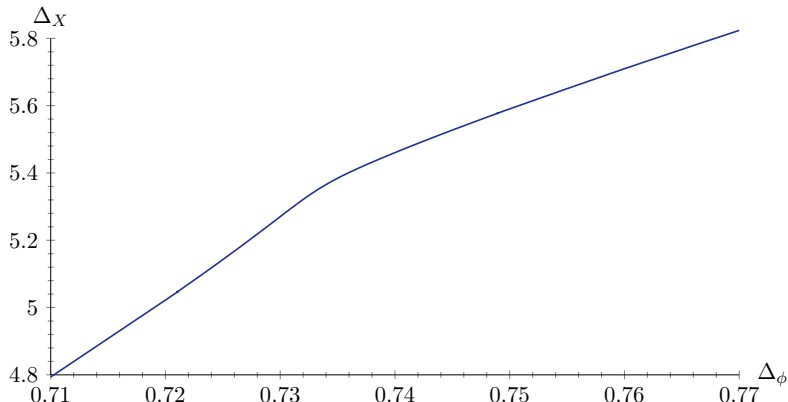

Figure 10: Single-correlator $MN_{20,2}$ exclusion bound. The line corresponds to the maximum allowed scaling dimension of the first scalar $X$ operator as a function of the scaling dimension of the order parameter operator. In qboot we used $\Lambda = 45$, $\ell = \{0, \ldots, 60, 63, 64, 66, 67, 73, 74, 77, 78, 81, 82, 85, 86, 89, 90, 93, 94, 97, 98\}$ and $\nu_{\max} = 26$.

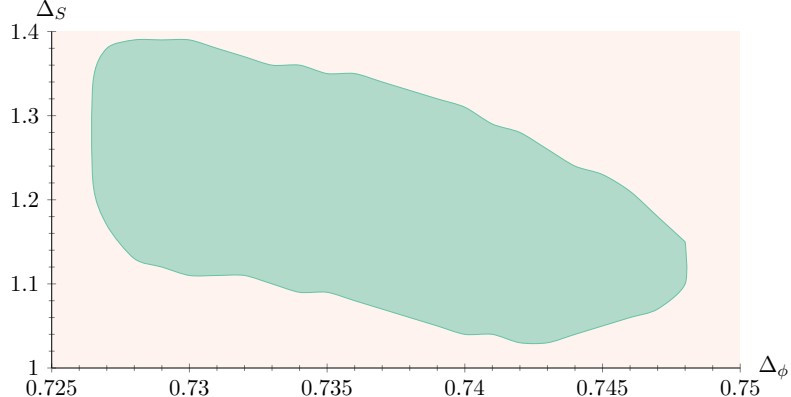

Figure 11: $MN_{20,2}$ island in the $\phi$-$S$ plane of parameter space. To obtain this figure we assumed that the first spin-two singlet is the stress-energy tensor and that $X$ saturates the bound of Fig. 10. We further imposed $\Delta_{S'} \geqslant 3.0$, $\Delta_{T'_{\mu\nu}} \geqslant 6.0$ (next-to-leading spin-two singlet), $\Delta_{X'} \geqslant 10.0$, $\Delta_{\phi'} \geqslant 1.0$. In qboot we used $\Lambda = 35$, $\ell = \{0, \ldots, 50, 55, 56, 59, 60, 64, 65, 69, 70, 74, 75, 79, 80, 84, 85, 89, 90\}$ and $\nu_{\max} = 25$.

other CFT data agreed exceptionally well with the perturbative predictions. We believe that sensitivity to the second scalar singlet should be restored once $S$ is included as an external operator.

## 3.3 The $n = 2$ $Z_{ij}^{ab} \times Z_{kl}^{cd}$ single correlator bootstrap

In Fig. 12 we display the bound for the maximum allowed scaling dimension for the first operator in the $SY$ representation as a function of $\Delta_Z$. For this plot we use the sum rules from Appendix B. There seems to be a kink roughly around the position of the second $SY$ operator and first $Z$ operator of the corresponding $MN_{100,2}$ theory studied in [25]. Beyond this similarity at the level of certain scaling dimensions we do not know if this kink is due to the same theory or some other theory. This is in part due to the fact that [25, Fig. 4] converges very slowly with $\Lambda$.

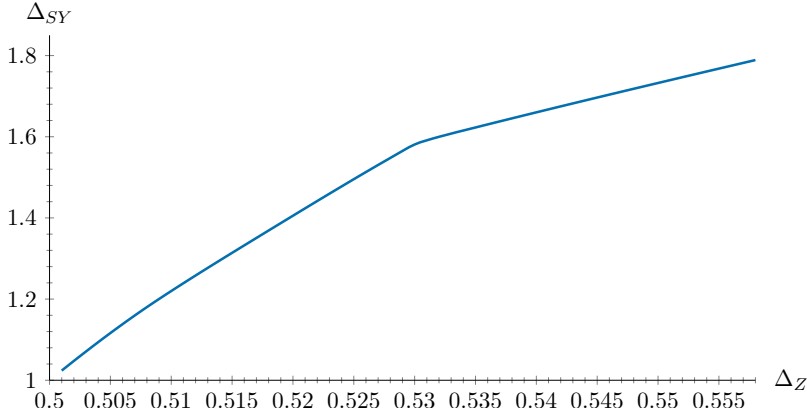

Figure 12: Single correlator $MN_{100,2}$ exclusion bound using the $\langle ZZZZ \rangle$ crossing equation. The line corresponds to the maximum allowed scaling dimension of the first $SY$ (remember $SY = Y$) operator as a function of the scaling dimension of the order parameter operator. To obtain this figure with qboot we used $\Lambda = 20$, $\ell = \{0, \ldots, 50, 55, 56, 59, 60, 64, 65, 69, 70, 74, 75, 79, 80, 84, 85, 89, 90\}$ and $\nu_{\max} = 25$.

## 3.4 The $n \geqslant 4$ $Z_{ij}^{ab} \times Z_{kl}^{cd}$ single correlator bootstrap

With the expressions for the sum rules of a four-point function of $Z_{ij}^{ab}$ operators worked out for generic $m \geqslant 2$ and $n \geqslant 4$, it is interesting to study the behavior of the bounds in various parameter limits. To this end, in Fig. 13 we probe the large $m$ limit, whereas in Fig. 14 we probe the large $n$ limit. We find that the most interesting exclusion bound is the one in the $SY$ sector, which displays a sharp kink for all values of $m$ and $n$ tested. In the large $n$ limit the kink converges to the point ($\Delta_{SY} = 2$, $\Delta_\phi = 1$) which hints at a "standard"[17] large $n$ Hubbard–Stratonovich description. On the other hand, it is not clear what the theory converges to in the large $m$ limit.

Another interesting thing to look at is the exclusion bound for the first operator transforming in the $BB$ representation. Representations that are antisymmetric in two pairs of indices tend to require higher powers of the field or derivatives in order to be written down and not vanish identically;[18] see related discussions in [49] and [36]. We note that this is true for theories where the fields may be written as polynomials of other fields, i.e. which have a weakly coupled description. However, we do expect intuitively that some qualitative features may carry over to the strong coupling limit. This is indeed what we observe in Fig. 15 where we find a kink that has (very roughly) $\Delta_{BB} \sim 2\Delta_Z + 2 \sim 4\Delta_Z$.

Let us note that we have checked that our bounds do not change if we assume that the external $Z$ operator appears in its OPE with itself. We checked this by looking at the $\Delta_{SZ} \sim \Delta_Z$ exclusion bound, for e.g. $m = 5$ and $n = 5$ and did not see any difference in our bounds (up to a vertical precision of $10^{-6}$ that we checked). More explicitly, we checked that if $Z$ is exchanged in the $Z \times Z$ OPE, then the exclusion bound on the second exchanged operator in this representation is identical to the exclusion bound of the first exchanged operator in the $Z$ irrep if one assumes that the external operator is not exchanged. Additionally, we checked that for e.g. $m = 1000$ and $n = 4$ the corresponding $\Delta_{SY}$ exclusion bound in Fig. 13 remains unchanged even after adding the assumption $\Delta_{SZ} \geqslant 1.0$. The question of whether or not $Z$ should appear in its OPE with itself is important, since if it does not, that would exclude

---

[17]For the "non-standard" large $n$ description see the discussion in section 3.1.1

[18]We thank Alessandro Vichi for pointing us in this direction.

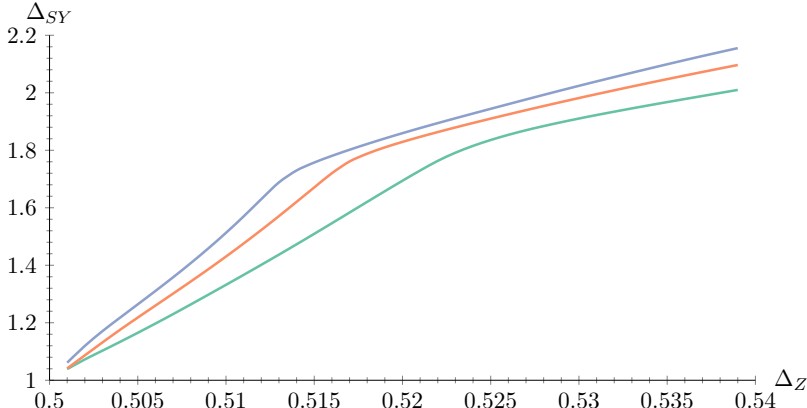

Figure 13: Single correlator MN exclusion bound using the $n \geqslant 4$ $\langle ZZZZ \rangle$ crossing equation. We display the behavior for $n = 4$ fixed and for increasing $m$ ($m = 10$ green, $m = 100$ red and $m = 1000$ blue). The line corresponds to the maximum allowed scaling dimension of the first $SY$ (remember $SY = Y$) operator as a function of the scaling dimension of the order parameter operator. To obtain this figure with qboot we used $\Lambda = 15$, $\ell = \{0, \dots, 50, 55, 56, 59, 60, 64, 65, 69, 70, 74, 75, 79, 80, 84, 85, 89, 90\}$ and $\nu_{\max} = 25$.

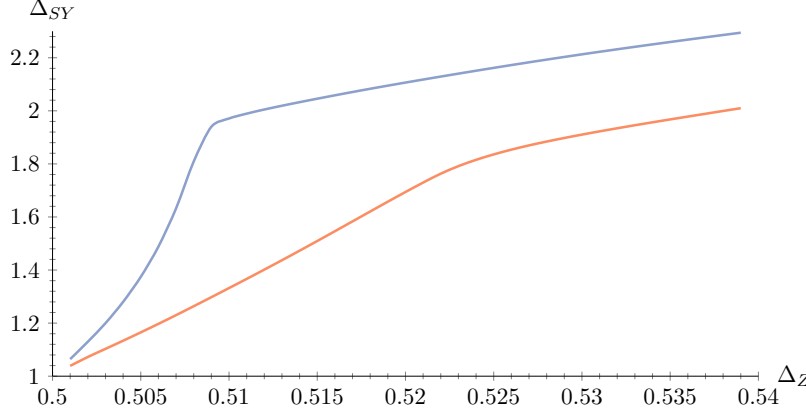

Figure 14: Single correlator MN exclusion bound using the $n \geqslant 4$ $\langle ZZZZ \rangle$ crossing equation. We display the behavior for $m = 10$ fixed and for $n = 4$ (red) and $n = 10$ (blue). The line corresponds to the maximum allowed scaling dimension of the first $SY$ (remember $SY = Y$) operator as a function of the scaling dimension of the order parameter operator. To obtain this figure with qboot we used $\Lambda = 15$, $\ell = \{0, \dots, 50, 55, 56, 59, 60, 64, 65, 69, 70, 74, 75, 79, 80, 84, 85, 89, 90\}$ and $\nu_{\max} = 25$.

a cubic (in powers of the order parameter field) term in a possible Hamiltonian/Lagrangian description of the theory. This is because in this case the $Z$ field would transform under an additional $\mathbb{Z}_2$ symmetry.

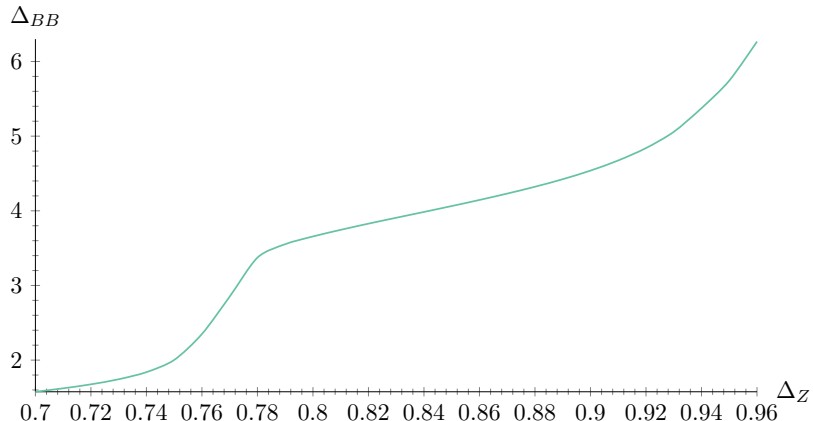

Figure 15: Single correlator MN exclusion bound using the $n = 10$ and $m = 2$ $\langle ZZZZ \rangle$ crossing equation. The line corresponds to the maximum allowed scaling dimension of the first $BB$ operator in the $Z \times Z$ OPE. To obtain this figure with qboot we used $\Lambda = 15$, $\ell = \{0, \ldots, 50, 55, 56, 59, 60, 64, 65, 69, 70, 74, 75, 79, 80, 84, 85, 89, 90\}$ and $\nu_{\max} = 25$.

## 4  Summary and conclusion

We have studied kinks and islands that arise when three-dimensional CFTs with $O(m)^n \rtimes S_n$ global symmetry are analyzed with the numerical bootstrap both "close" and "far" from the unitarity bound. With regards to the analysis close to the unitarity bound, we found that in the large number of copies limit the kinks converge to the expected position from perturbative calculations. However, when we examine theories with a small number of copies we do not agree with the $\varepsilon$ expansion predictions. This has two interpretations. The first interpretation is that the theories we find are those of the $\varepsilon$ expansion, but for some reason the actual perturbative predictions are very inaccurate. The second is that the kinks are due to another theory that converges to the same point when the number of copies is taken to be large. Note that this is precisely the same qualitative behavior observed in the case of hypercubic theories.

In addition to the above, we have worked out the tensor structures for problems including groups of the form $G^n \rtimes S_n$ with $G$ and $n$ arbitrary. To this end, as an application of these results we bootstrapped the four-point function of symmetric tensors $Z_{ij}^{ab}$ of $O(m)^n \rtimes S_n$. Although we found various interesting features, we left a more detailed analysis of these theories for future work.

There are various future directions that stem from the present work. One is to bootstrap a mixed system of correlators involving $\phi_i^a$ and $Z_{jk}^{bc}$, this would give us a better handle on theories like the ones in Figs. 11 and 12. Also, having worked out the tensor structures for $G^n \rtimes S_n$ it would be interesting to bootstrap the theories consisting of $n$ copies of the $m$ state Potts models and see if we can find any interesting features; see e.g. [52].

## Acknowledgements

We thank Mocho Go for assistance regarding the implementation of OPE coefficient relations and definitions in qboot. We are grateful to Alessandro Vichi for useful discussions and suggestions and to Johan Henriksson for reading through the manuscript and providing useful comments. Additionally, we thank two anonymous referees for their careful reading and constructive comments that helped improve this manuscript. Numerical computations in this

paper were run on the Crete Center for Quantum Complexity and Nanotechnology. This research used resources provided by the Los Alamos National Laboratory Institutional Computing Program, which is supported by the U.S. Department of Energy National Nuclear Security Administration under Contract No. 89233218CNA000001. This research used resources of the National Energy Research Scientific Computing Center (NERSC), a U.S. Department of Energy Office of Science User Facility operated under Contract No. DE-AC02-05CH11231.

The research work of SRK was supported by the Hellenic Foundation for Research and Innovation (HFRI) under the HFRI PhD Fellowship grant (Fellowship Number: 1026). The research work of SRK also received funding from the European Research Council (ERC) under the European Union's Horizon 2020 research and innovation programme (grant agreement no. 758903).

Research presented in this article was supported by the Laboratory Directed Research and Development program of Los Alamos National Laboratory under project number 20180709PRD1. AS is funded by the Royal Society under the grant "Advancing the Conformal Bootstrap Program in Three and Four Dimensions".

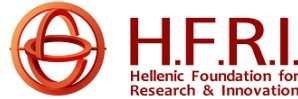

# A $\phi$-X system sum rules

In this appendix we collect the sum rules that result from each crossing equation in the system of $\phi$-X mixed correlator equations. We define $F_{\pm,\Delta,l}^{ij,kl}$ as in [53] and we also use $F^{\pm}$ and $F_{\pm}$ interchangeably.

## A.1 $\langle \phi\phi\phi\phi \rangle$ crossing equation

The single correlator sum rules have already appeared in [24] and [25]. We quote them below for completeness:

$$
\sum_{S^+} c_O^2 \begin{pmatrix} 0 \\ F_{\Delta,\ell}^- \\ 0 \\ 0 \\ F_{\Delta,\ell}^+ \\ 0 \end{pmatrix}
+ \sum_{X^+} c_O^2 \begin{pmatrix} 0 \\ -F_{\Delta,\ell}^- \\ nF_{\Delta,\ell}^- \\ 0 \\ -F_{\Delta,\ell}^+ \\ nF_{\Delta,\ell}^+ \end{pmatrix}
+ \sum_{Y^+} c_O^2 \begin{pmatrix} 0 \\ 0 \\ \frac{m-2}{2}F_{\Delta,\ell}^- \\ mF_{\Delta,\ell}^- \\ 0 \\ \frac{m-2}{2}F_{\Delta,\ell}^+ \end{pmatrix}
+ \sum_{Z^+} c_O^2 \begin{pmatrix} F_{\Delta,\ell}^- \\ \frac{1}{2}F_{\Delta,\ell}^- \\ -\frac{1}{2}F_{\Delta,\ell}^- \\ -F_{\Delta,\ell}^- \\ -\frac{1}{2}F_{\Delta,\ell}^+ \\ +\frac{1}{2}F_{\Delta,\ell}^+ \end{pmatrix}
$$

$$
+ \sum_{A^-} c_O^2 \begin{pmatrix} 0 \\ 0 \\ -\frac{1}{2}F_{\Delta,\ell}^- \\ F_{\Delta,\ell}^- \\ 0 \\ \frac{1}{2}F_{\Delta,\ell}^+ \end{pmatrix}
+ \sum_{B^-} c_O^2 \begin{pmatrix} F_{\Delta,\ell}^- \\ -\frac{1}{2}F_{\Delta,\ell}^- \\ \frac{1}{2}F_{\Delta,\ell}^- \\ -F_{\Delta,\ell}^- \\ \frac{1}{2}F_{\Delta,\ell}^+ \\ -\frac{1}{2}F_{\Delta,\ell}^+ \end{pmatrix}
= \begin{pmatrix} 0 \\ 0 \\ 0 \\ 0 \\ 0 \\ 0 \end{pmatrix}.
\tag{25}
$$

## A.2  $\langle \phi X \phi X \rangle$ crossing equation

Using the projectors (11) and (12) we find the sum rules

$$\sum_O \lambda_{\phi X O_{\bar{A}}}^2 F_{\mp,\Delta,l}^{\phi X,\phi X} = 0, \tag{26}$$

and

$$\sum_O \lambda_{\phi X O_\phi}^2 F_{\mp,\Delta,l}^{\phi X,\phi X} = 0. \tag{27}$$

## A.3  $\langle \phi \phi X X \rangle$ crossing equation

For the crossing equation from the four-point function $\langle \phi \phi X X \rangle$ we also need the projectors corresponding to the decomposition of $\langle \phi_i^a \phi_j^b X^{cd} X^{ef} \rangle$ onto irreps. To find these we need to know the irreps the $\phi \times \phi$ and $X \times X$ OPEs have in common. These are the $X$ and $S$ irreps. The corresponding projectors turn out to be [13, 34]

$$P_{abcdef}^S = \delta_{ab}\Big(\delta_{cdef} - \frac{1}{n}\delta_{cd}\delta_{ef}\Big) \tag{28}$$

and

$$P_{abcdef}^X = \delta_{abcdef} - \frac{1}{n}\Big(\delta_{ab}\delta_{cdef} + \delta_{cd}\delta_{abef} + \delta_{ef}\delta_{abcd}\Big) + \frac{2}{n^2}\delta_{ab}\delta_{cd}\delta_{ef}. \tag{29}$$

We thus obtain the sum rules

$$\sum_O \big(\lambda_{\phi \phi O_{S+}} \lambda_{X X O_{S+}} F_{\mp,\Delta,l}^{\phi \phi,XX} \pm (-1)^l \frac{1}{n}\lambda_{\phi X O_y}^2 F_{\mp,\Delta,l}^{X\phi,\phi X} \pm (-1)^l \frac{1}{n}\lambda_{\phi X O_{\bar{A}}}^2 F_{\mp,\Delta,l}^{X\phi,\phi X}\big) = 0 \tag{30}$$

and

$$\sum_O \big(\lambda_{\phi \phi O_{X+}} \lambda_{X X O_{X+}} F_{\mp,\Delta,l}^{\phi \phi,XX} \pm (-1)^l \lambda_{\phi X O_y}^2 F_{\mp,\Delta,l}^{X\phi,\phi X} \mp (-1)^l \lambda_{\phi X O_{\bar{A}}}^2 F_{\mp,\Delta,l}^{X\phi,\phi X}\big) = 0. \tag{31}$$

## A.4  $\langle X X X X \rangle$ crossing equation

For the $\langle X X X X \rangle$ crossing equation the sum rules have been already computed in the literature, albeit in a slightly different context, see e.g. [11] and [12]. We quote

$$\sum_{S^+} c_O^2 \begin{pmatrix} 0 \\ F_{\Delta,\ell}^- \\ F_{\Delta,\ell}^+ \\ 0 \end{pmatrix} + \sum_{X^+} c_O^2 \begin{pmatrix} 0 \\ 0 \\ -\frac{4}{n+1}F_{\Delta,\ell}^+ \\ F_{\Delta,\ell}^- \end{pmatrix} + \sum_{E^+} c_O^2 \begin{pmatrix} F_{\Delta,\ell}^- \\ \frac{2(n-1)}{n}F_{\Delta,\ell}^- \\ -\frac{(n+1)(n-2)}{n(n-1)}F_{\Delta,\ell}^+ \\ -\frac{n+1}{2(n-1)}F_{\Delta,\ell}^- \end{pmatrix} + \sum_{\bar{S}^-} c_O^2 \begin{pmatrix} F_{\Delta,\ell}^- \\ 0 \\ F_{\Delta,\ell}^+ \\ 0 \end{pmatrix} = \begin{pmatrix} 0 \\ 0 \\ 0 \\ 0 \end{pmatrix}. \tag{32}$$

### A.4.1  The $n = 3$ case

For $n = 3$ the sum rules become

$$\sum_{S^+} c_O^2 \begin{pmatrix} 0 \\ F_{\Delta,\ell}^- \\ F_{\Delta,\ell}^+ \end{pmatrix} + \sum_{X^+} c_O^2 \begin{pmatrix} F_{\Delta,\ell}^- \\ 0 \\ -2F_{\Delta,\ell}^+ \end{pmatrix} + \sum_{\bar{S}^-} c_O^2 \begin{pmatrix} F_{\Delta,\ell}^- \\ -F_{\Delta,\ell}^- \\ F_{\Delta,\ell}^+ \end{pmatrix} = \begin{pmatrix} 0 \\ 0 \\ 0 \end{pmatrix},$$

which are equivalent to those of $O(2)$.

## A.5  $n = 2$ sum rules

With the exception of the single correlator sum rules, all the above sum rules are drastically simplified when $n = 2$, this is because the irrep $X$ becomes one dimensional then.[19] We thus

---

[19]Hence, all OPEs involving $X$ only exchange one irrep.

have

$$\sum_{S^+} \lambda_{XXO_S}^2 F_{\Delta,l}^{-XX,XX} = 0 \,, \tag{33}$$

$$\sum_{\phi^\pm} \lambda_{OXO_\phi}^2 F_{\Delta,l}^{-OX,OX} = 0 \,, \tag{34}$$

$$\sum_{S^+} \lambda_{OOO_S} \lambda_{XXO_S} F_{\Delta,l}^{\mp OO,XX} \pm \sum_{\phi^\pm} (-1)^l \lambda_{OXO_\phi}^2 F_{\Delta,l}^{\mp XO,OX} = 0 \,, \tag{35}$$

in addition to the single correlator sum rules which remain as defined earlier.

## B   $Z \times Z$ single correlator sum rules

The $n = 2$ $Z \times Z$ bootstrap the sum rules are simplified drastically by the observation that $Z_{ij}^{ab}$ and $Z_{kl}^{cd}$ necessarily have ($a = c$ and $b = d$) or ($a = d$ and $b = c$). Thus if we assume that the field $Z_{ij}^{ab}$ has, lets say, $a = 1$ then by definition $b = 2$. Hence, the projectors are

$$
\begin{aligned}
P^S_{ijklmnop} &= P^S_{ikmo} P^S_{jlnp} \,, \\
P^{SY}_{ijklmnop} &= P^S_{ikmo} P^Y_{jlnp} + P^Y_{ikmo} P^S_{jlnp} \,, \\
P^{SA}_{ijklmnop} &= P^S_{ikmo} P^A_{jlnp} + P^A_{ikmo} P^S_{jlnp} \,, \\
P^{YA}_{ijklmnop} &= P^Y_{ikmo} P^A_{jlnp} + P^A_{ikmo} P^Y_{jlnp} \,, \\
P^{YY}_{ijklmnop} &= P^Y_{ikmo} P^Y_{jlnp} \,, \\
P^{AA}_{ijklmnop} &= P^A_{ikmo} P^A_{jlnp} \,,
\end{aligned}
\tag{36}
$$

i.e. the products of projectors of $O(m)$. By $S$ we denote the scalar singlet, by $A$ the antisymmetric and by $Y$ the rank-two traceless symmetric irreps of $O(m)$. We obtain the following sum rules

$$
\begin{aligned}
&\sum_{S^+} c_O^2 \begin{pmatrix} 0 \\ F_{\Delta,\ell}^- \\ 0 \\ 0 \\ 0 \\ F_{\Delta,\ell}^+ \end{pmatrix}
+ \sum_{SY^+} c_O^2 \begin{pmatrix} 0 \\ -2F_{\Delta,\ell}^- \\ F_{\Delta,\ell}^- \\ 2F_{\Delta,\ell}^- \\ -2F_{\Delta,\ell}^+ \\ \frac{m-4}{2} F_{\Delta,\ell}^+ \end{pmatrix}
+ \sum_{YY^+} c_O^2 \begin{pmatrix} F_{\Delta,\ell}^- \\ F_{\Delta,\ell}^- \\ -F_{\Delta,\ell}^- \\ (m-2)F_{\Delta,\ell}^- \\ (m+2)F_{\Delta,\ell}^+ \\ \frac{2-m-m^2}{2} F_{\Delta,\ell}^+ \end{pmatrix}
+ \sum_{AA^+} c_O^2 \begin{pmatrix} F_{\Delta,\ell}^- \\ 0 \\ 0 \\ 0 \\ -m F_{\Delta,\ell}^+ \\ 0 \end{pmatrix} \\
&\qquad + \sum_{SA^-} c_O^2 \begin{pmatrix} 0 \\ 0 \\ -F_{\Delta,\ell}^- \\ 0 \\ -2F_{\Delta,\ell}^+ \\ \frac{m}{2} F_{\Delta,\ell}^+ \end{pmatrix}
+ \sum_{YA^-} c_O^2 \begin{pmatrix} 2F_{\Delta,\ell}^- \\ -m^2 F_{\Delta,\ell}^- \\ F_{\Delta,\ell}^- \\ m F_{\Delta,\ell}^- \\ 2F_{\Delta,\ell}^+ \\ \frac{m^2-m}{2} F_{\Delta,\ell}^+ \end{pmatrix}
= \begin{pmatrix} 0 \\ 0 \\ 0 \\ 0 \\ 0 \\ 0 \end{pmatrix} .
\end{aligned}
\tag{37}
$$

# C  $\langle ZZZZ \rangle$ projectors for $n \geqslant 4$ and $G$ arbitrary

The projectors that correspond to the $\langle Z_{ij}^{ab} Z_{kl}^{cd} Z_{mn}^{ef} Z_{op}^{gh} \rangle$ correlator are rather extended 16 index objects. In order to simplify their presentation we will introduce "Blocks", not to be confused with conformal blocks. With the use of Blocks the projectors can be presented in somewhat compact form. Also, we will make no distinction between upper and lower indices. It will be implicitly assumed that indices $a$-$h$ label the copy of $G$, and the indices $i$-$p$ are $G$ indices. This setup is not the most general possible with respect to $G$ indices, but we use it for simplicity of demonstration. We hope that the generalization to arbitrary indices of $G$ will become obvious from our presentation. There are three groups of representations that appear in the $Z_{ij}^{ab} \times Z_{kl}^{cd}$ OPE. These are:

Group I: Representations with $a = c$ or $d$ OR $b = c$ or $d$.

Group II: Representations with $a = c$ or $d$ AND $b = d$ or $c$.

Group III: Representations with $a$, $b$, $c$ and $d$ all different.

To reiterate, we have Group I: two pairs of indices equal, Group II: one pair of indices equal, Group III: no pairs of indices equal. We also remind the reader that the operators $Z_{ij}^{ab}$ have $a \neq b$.

## C.1  Group I representations

Let us start by recalling the projectors of the $G$ irreps (we take $G = O(m)$ in order to be explicit, but the generalization to any $G$ is trivial). We have $P_{g_1}^{ijkl} = \frac{1}{m}\delta^{ij}\delta^{kl}$, $P_{g_2}^{ijkl} = \frac{1}{2}(\delta^{ik}\delta^{jl} - \delta^{il}\delta^{jk})$ and $P_{g_3}^{ijkl} = \frac{1}{2}(\delta^{ik}\delta^{jl} + \delta^{il}\delta^{jk} - \frac{2}{m}\delta^{ij}\delta^{kl})$. Next we define the following useful tensors:

$$\begin{aligned}
R_{1abcdefgh} &= (\delta_{ac}\delta_{bd} - \delta_{acbd})(\delta_{eg}\delta_{fh} - \delta_{egfh}), \\
R_{2abcdefgh} &= \delta_{aceg}\delta_{bd}\delta_{fh} - \delta_{abcdeg}\delta_{fh} - \delta_{afcheg}\delta_{bd} + \delta_{abcdefgh}, \\
R_{3abcdefgh} &= \delta_{aceg}\delta_{bdfh} - \delta_{abcdefgh}.
\end{aligned} \tag{38}$$

With these tensors in hand we can now define the Blocks. We denote these with "$B$" for short. They are

$$\begin{aligned}
B_{S\,ijklmnop}^{\phantom{S}abcdefgh} &= P_{g_1}^{ikmo} P_{g_1}^{jlnp} R_{1abcdefgh}, \\
B_{XS\,ijklmnop}^{\phantom{XS}abcdefgh} &= P_{g_1}^{ikmo} P_{g_1}^{jlnp} \left( R_{2abcdefgh} - \frac{1}{n} R_{1abcdefgh} \right), \\
B_{XY\,ijklmnop}^{\phantom{XY}abcdefgh} &= P_{g_1}^{ikmo} P_{g_3}^{jlnp} \left( R_{3abcdefgh} - \frac{1}{n-1} R_{2badcfehg} \right), \\
B_{XA\,ijklmnop}^{\phantom{XA}abcdefgh} &= P_{g_1}^{ikmo} P_{g_2}^{jlnp} \left( R_{3abcdefgh} - \frac{1}{n-1} R_{2badcfehg} \right), \\
B_{SY\,ijklmnop}^{\phantom{SY}abcdefgh} &= P_{g_1}^{ikmo} P_{g_3}^{jlnp} R_{2badcfehg}, \\
B_{SA\,ijklmnop}^{\phantom{SA}abcdefgh} &= P_{g_1}^{ikmo} P_{g_2}^{jlnp} R_{2badcfehg}, \\
B_{YY\,ijklmnop}^{\phantom{YY}abcdefgh} &= P_{g_3}^{ikmo} P_{g_3}^{jlnp} R_{3badcfehg}, \\
B_{AA\,ijklmnop}^{\phantom{AA}abcdefgh} &= P_{g_2}^{ikmo} P_{g_2}^{jlnp} R_{3badcfehg}, \\
B_{YA\,ijklmnop}^{\phantom{YA}abcdefgh} &= P_{g_3}^{ikmo} P_{g_2}^{jlnp} R_{3badcfehg},
\end{aligned} \tag{39}$$

$$B_{\bar{X}\,ijklmnop}^{abcdefgh} = P_{g_1}^{ikmo} P_{g_1}^{jlnp}\Big(R_{3\,badcfehg} - \frac{1}{n-2}(R_{2\,abcdefgh} + R_{2\,badcfehg})$$
$$+ \frac{1}{(n-1)(n-2)}R_{1\,abcdefgh}\Big).$$

To finally obtain the projectors from the Blocks we must perform symmetrizations, which are the same for all Blocks, hence we write $B_g$, where $g$ is one of the irreps considered above. These symmetrizations are

$$B_g'^{\,abcdefgh}_{\ ijklmnop} = B_g^{\,abcdefgh}_{\ ijklmnop} + B_g^{\,abcdefhg}_{\ ijklmnpo}\,,$$
$$B_g''^{\,abcdefgh}_{\ ijklmnop} = B_g'^{\,abcdefgh}_{\ ijklmnop} + B_g'^{\,abcdfegh}_{\ ijklnmop}\,,$$
$$B_g'''^{\,abcdefgh}_{\ ijklmnop} = B_g''^{\,abcdefgh}_{\ ijklmnop} + B_g''^{\,abdcef,gh}_{\ ijlkmnop}\,,$$
$$P_g^{\,abcdefgh}_{\ ijklmnop} = B_g'''^{\,abcdefgh}_{\ ijklmnop} + B_g'''^{\,bacdefgh}_{\ jiklmnop}\,,$$

(40)

where the last line in (40) corresponds to the final expression for the projector in irrep $g$. Note that the above symmetrizations are necessary in order to take into account the symmetry $Z_{ij}^{ab} = Z_{ji}^{ba}$. To apply these equations to a different group $G$ one simply needs to replace the expressions $P_{g_1}$, $P_{g_2}$ and $P_{g_3}$ with those of their group of choice.

## C.2  Group II representations

For Group II representations the steps are very similar with the ones described for Group I. We must simply define some new tensors. These are

$$RR_{1\,abcdefgh} = \delta_{aceg}(\delta_{bf}\delta_{dh} - \delta_{bfdh}) - (\delta_{acegbf}\delta_{dh} - \delta_{acegbfdh}) - (\delta_{acegdh}\delta_{bf} - \delta_{acegbfdh})$$
$$RR_{2\,abcdefgh} = \delta_{ac}\delta_{eg}(\delta_{bf}\delta_{dh} - \delta_{bfdh}) - \delta_{eg}(\delta_{acbf}\delta_{dh} + \delta_{acdh}\delta_{bf} - 2\delta_{acbfdh}),$$
$$\quad - \delta_{ac}(\delta_{egbf}\delta_{dh} + \delta_{egdh}\delta_{bf} - 2\delta_{egbfdh}) + (\delta_{acbfeg}\delta_{dh} + \delta_{acdheg}\delta_{bf} - 2\delta_{acegbfdh})$$
$$\quad + (\delta_{acbf}\delta_{egdh} + \delta_{acdh}\delta_{bfeg} - 2\delta_{acegbfdh}),$$

(41)

whereas the Blocks for Group II representations are

$$B_{XB\,ijklmnop}^{abcdefgh} = (RR_{1\,abcdefgh} - \frac{1}{n-2}RR_{2\,abcdefgh})P_{g_1}^{ikmo}\delta_{jn}\delta_{lp}\,,$$
$$\quad - (RR_{1\,abcdehgf} - \frac{1}{n-2}RR_{2\,abcdehgf})P_{g_1}^{ikmo}\delta_{jp}\delta_{ln}\,,$$
$$B_{XZ\,ijklmnop}^{abcdefgh} = (RR_{1\,abcdefgh} - \frac{1}{n-2}RR_{2\,abcdefgh})P_{g_1}^{ikmo}\delta_{jn}\delta_{lp}\,,$$
$$\quad + (RR_{1\,abcdehgf} - \frac{1}{n-2}RR_{2\,abcdehgf})P_{g_1}^{ikmo}\delta_{jp}\delta_{ln}\,,$$
$$B_{SB\,ijklmnop}^{abcdefgh} = RR_{2\,abcdefgh}P_{g_1}^{ikmo}\delta_{jn}\delta_{lp} - RR_{2\,abcdehgf}P_{g_1}^{ikmo}\delta_{jp}\delta_{ln}\,,$$
$$B_{SZ\,ijklmnop}^{abcdefgh} = RR_{2\,abcdefgh}P_{g_1}^{ikmo}\delta_{jn}\delta_{lp} + RR_{2\,abcdehgf}P_{g_1}^{ikmo}\delta_{jp}\delta_{ln}\,,$$
$$B_{YB\,ijklmnop}^{abcdefgh} = RR_{1\,abcdefgh}P_{g_3}^{ikmo}\delta_{jn}\delta_{lp} - RR_{1\,abcdehgf}P_{g_3}^{ikmo}\delta_{jp}\delta_{ln}\,,$$
$$B_{YZ\,ijklmnop}^{abcdefgh} = RR_{1\,abcdefgh}P_{g_3}^{ikmo}\delta_{jn}\delta_{lp} + RR_{1\,abcdehgf}P_{g_3}^{ikmo}\delta_{jp}\delta_{ln}\,,$$
$$B_{AB\,ijklmnop}^{abcdefgh} = RR_{1\,abcdefgh}P_{g_2}^{ikmo}\delta_{jn}\delta_{lp} - RR_{1\,abcdehgf}P_{g_2}^{ikmo}\delta_{jp}\delta_{ln}\,,$$
$$B_{AZ\,ijklmnop}^{abcdefgh} = RR_{1\,abcdefgh}P_{g_2}^{ikmo}\delta_{jn}\delta_{lp} + RR_{1\,abcdehgf}P_{g_2}^{ikmo}\delta_{jp}\delta_{ln}\,.$$

(42)

We may now apply (40) and obtain the final expression for the projectors.

## C.3 Group III representations

For the last group of representations we need to define one more tensor structure:

$$
\begin{aligned}
R_{4\,ijklmnop}^{\;abcdefgh} = (&\delta_{ae}\delta_{bf}\delta_{cg}\delta_{dh} - \delta_{cg}\delta_{dh}\delta_{abef} - \delta_{bf}\delta_{dh}\delta_{aceg} - \delta_{bf}\delta_{cg}\delta_{adeh} - \delta_{ae}\delta_{dh}\delta_{bcfg} \\
&- \delta_{ae}\delta_{cg}\delta_{bdfh} - \delta_{ae}\delta_{bf}\delta_{cdgh} + \delta_{aedh}\delta_{bcfg} + \delta_{aecg}\delta_{bdfh} + \delta_{aebf}\delta_{cdgh} \\
&+ 2(\delta_{dh}\delta_{abcefg} + \delta_{cg}\delta_{abdefh} + \delta_{bf}\delta_{adcehg} + \delta_{ae}\delta_{dbchfg}) - 6\delta_{aecgbfdh}) \\
&\times \delta_{im}\delta_{ko}\delta_{jn}\delta_{lp}\,,
\end{aligned}
\tag{43}
$$

leading to the Blocks

$$
\begin{aligned}
B_{BB\,ijklmnop}^{\;\;abcdefgh} &= R_{4\,ijklmnop}^{\;abcdefgh} - R_{4\,ijklonmp}^{\;abcdgfeh} - R_{4\,ijklmpon}^{\;abcdehgf} + R_{4\,ijklopmn}^{\;abcdghef}\,, \\
B_{BZ\,ijklmnop}^{\;\;abcdefgh} &= R_{4\,ijklmnop}^{\;abcdefgh} - R_{4\,ijklonmp}^{\;abcdgfeh} + R_{4\,ijklmpon}^{\;abcdehgf} - R_{4\,ijklopmn}^{\;abcdghef}\,, \\
B_{TotS\,ijklmnop}^{\;\;\;abcdefgh} &= R_{4\,ijklmnop}^{\;abcdefgh} + R_{4\,ijklmnpo}^{\;abcdefhg}\,.
\end{aligned}
\tag{44}
$$

For the representations $BB$ and $BZ$ we may simply use (40) to obtain the projectors. On the other hand, $TotS$ needs a slightly more elaborate formula, which is due to our explicit choice of Block. We have

$$
\begin{aligned}
B_{TotS}^{\prime\,abcdefgh}{}_{ijklmnop} &= B_{TotS\,ijklmnop}^{\;\;\;abcdefgh} + B_{TotS\,ijklmonp}^{\;\;\;abcdegfh} + B_{TotS\,ijklmpon}^{\;\;\;abcdehgf}\,, \\
P_{TotS\,ijklmnop}^{\;\;\;abcdefgh} &= B_{TotS}^{\prime\,abcdefgh}{}_{ijklmnop} + B_{TotS}^{\prime\,abcdfegh}{}_{ijklnmop} + B_{TotS}^{\prime\,abcdgfeh}{}_{ijklonmp} + B_{TotS}^{\prime\,abcdhfge}{}_{ijklpnom}\,.
\end{aligned}
\tag{45}
$$

## C.4 Comments

The above tensors can be derived if we know the symmetry properties of each irrep (symmetric, antisymmetric, traceless, trace, ...), by writing down a tensor with the same symmetry and then contracting with a tensor that enforces the constraints $a \neq b$, $c \neq d$, $e \neq f$ and $g \neq h$. Such a tensor can be found in a straightforward way. First, define a reduced Kronecker delta $\delta_{ab}^r = \delta_{ab} - \delta_{abr}$, which is the same as the usual Kronecker delta, but equal to zero if the indices are equal to a value $r$. Now we can define $P_{a'b'c'd'}^{abcd} = \delta_{aa'}^b \delta_{bb'}\delta_{cc'}^d \delta_{dd'}$ which is simplified as $P_{a'b'c'd'}^{abcd} = (\delta_{aa'}\delta_{bb'} - \delta_{aa'bb'})(\delta_{cc'}\delta_{dd'} - \delta_{cc'dd'})$ by plugging in the definition of the reduced Kronecker delta. Now suppose that we had found a tensor $T^{a'b'c'd'e'f'g'h'}$ with all the required symmetry properties to describe some irrep. We could turn it into a projector using the following equation:

$$
T^{abcdefgh} = P_{a'b'c'd'}^{abcd}P_{e'f'g'h'}^{efgh}T^{a'b'c'd'e'f'g'h'}\,,
\tag{46}
$$

where the left-hand side is our projector. One can also define generalized reduced Kronecker deltas in order to impose more complicated constraints. For example, let us consider a tensor with four indices which must take different values and is totally symmetric (this is relevant for the $TotS$ irrep). We may start with a tensor $T^{a'b'c'd'}$ that is simply totally symmetric but with arbitrary index values. The constraint can then be imposed by contracting with $P_{a'b'c'd'}^{abcd} = \delta_{aa'}^{bcd}\delta_{bb'}^{cd}\delta_{cc'}^{d}\delta_{dd'}$. Note that $\delta_{bb'}^{cd} = \delta_{bb'} - \delta_{bb'c} - \delta_{bb'd} + \delta_{bb'dc}$. But the utility of this formalism is now clear: if one wishes to evaluate a relation such as $\delta_{bb'}^{cd}\delta_{cc'}^{d}\delta_{dd'}$, the last term in $\delta_{bb'}^{cd}$, namely $\delta_{bb'dc}$, can be dropped since it gives zero when contracted with $\delta_{cc'}^{d}\delta_{dd'} = (\delta_{cc'}\delta_{dd'} - \delta_{cc'dd'})$. Thus, in conclusion, as long as we calculate expressions like $\delta_{aa'}^{bcd}\delta_{bb'}^{cd}\delta_{cc'}^{d}\delta_{dd'}$ from right to left, it is sufficient to take $\delta_{bb'}^{a_1 a_2 \dots a_n} = \delta_{bb'} - \sum_{i=0}^{n}\delta_{bb'a_i}$. This allows us to algorithmically impose constraints on tensors.

# D $\langle ZZZZ \rangle$ sum rules for $n \geqslant 4$

Due to the rather extended size of the sum rules derived from the crossing equation corresponding to $\langle ZZZZ \rangle$ for $n \geqslant 4$ we attach them in an auxiliary file.

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
