# Peer review of "Bootstrapping Mixed MN Correlators in 3D"

_SciPost Physics, doi:SciPost Phys. 12, 206 (2022)_

## Round 1 · Referee Report · Anonymous (Referee 1) · 2022-2-28

Strengths

1-Novel work presented in a format that is instructive and useful to a reader within the field
2-Pedagogical presentation (such as going through examples of the $\phi_i^a\times\phi_j^b$ OPE) to make the work accessible
3-Lengthy and extensive appendices to document the intermediate technical results
4-General intermediate results that can be used in broader outside contexts (i.e., the $G^n\rtimes S_n$ OPE expansion results)

Weaknesses

1-Results themselves (the bounds and islands) are presented in a slightly confusing format that can become hard to follow
2-Some of the text could be better organized

Report

The manuscript presents novel results in bootstrapping the 3D $O(m)^n\rtimes S_n$ CFTs with mixed correlators. The results extend the conformal bootstrap's success in the $O(N)$ CFTs, which can be seen as the $n=1$ case of this manuscript's results. The authors find several rigorous bounds and less rigorous islands which will be of interest outside of the bootstrap community and leaves open room for further projects in this direction. Along the way, they derive more general OPE decompositions for operators transforming under representations of $G^n\rtimes S_n$, which will be useful for other bootstrap studies in future.

The authors do a good job of going through what amounts to fairly dense algebraic work in a way that is both useful and accessible to readers of varying technical expertise. Both an expert and an inexperienced student would be able to learn from section 2 especially. Additionally, the authors present multiple intermediate results in the appendix so that the various insights of this work can be used to not only reproduce but extend the work of this manuscript. In short, this manuscript makes a strong contribution to the broader conformal bootstrap project.

The main results of section 3 are impressive and robust and demonstrate a thorough understanding of what was studied. However, I found myself slightly confused due to the number of similar but distinct plots presented serially for the different models, as that hid the relations between the figures. I might suggest putting the $\Delta_\phi$-$\Delta_S$ and $\Delta_\phi$-$\Delta_X$ plots of the same model as subfigures within the same figure. In other words, have only one figure per model, with subfigures for each individual plot. Additionally, I had some trouble relating e.g. Figs 5 and 6 due to the changing axis scales: it was not immediately apparent to me that these were referring to the same parameter space. With this said, it was very interesting to have both of these plots back to back so as to show which "lobe" in fig 5 corresponded to which rough range of gap assumptions. I would suggest combining figs 5 and 6 into a single plot, if possible: I recognize that these are at different $\Lambda$, but perhaps an intersection plot would be a nice way of presenting this data.

Having done calculations of this sort, I can say that the ZZZZ single correlator work is heroic for the sheer volume of algebra that was successfully managed and presented. It's clear that the authors have spent a great deal of time and effort thinking through how best to present this dense subject matter. With this said, I found the exact placement of the results within the paper (in sections 3.3-5) to be slightly confusing; within the numerical results section, the authors spend a few pages on algebraic results that might have better fit in section 2 with the other OPE work. Relatedly, the numerical results subsection (3.5) within the numerical results section made it difficult for me when jumping around in the paper to find specific results. Besides moving the algebraic results to section 2, the authors might also consider organizing the paper with one or two sections for the $\phi$-$X$ systems along with their numerical results followed by another one or two sections for the $Z$ systems and their numerical results. I suspect it should be relatively easy to implement and would better showcase the excellent work done by the authors.

There were some minor grammatical or stylistic errors in the manuscript, but it is perfectly legible so it's not a problem.

In short, this manuscript is a useful, accessible, and important contribution to the field which will open new directions of research. With the above minor revisions, it achieves all of the criteria for acceptance.

Requested changes

1-Condense or otherwise reorganize the presentation of the plots so as to make clear which plots are related and which plots are distinct.
2-Modify sections 3.3-5 so as to prevent algebraic work from being presented in the midst of numerical work.

---

## Round 1 · Referee Report · Anonymous (Referee 2) · 2022-3-2

Strengths

  1. This paper advances the frontier of the numerical bootstrap by studying a set of crossing equations for MN models that have not previously been explored.
  2. The required new representation theory is explained well despite being quite complex.
  3. Details on the representation theory for a wider class of theories than studied in this paper are also worked out. This can be very useful for future numerical bootstrap studies.
  4. Characteristics of solutions to (truncated) crossing equations located close to previously discovered and new kinks in bounds are studied numerically. Under reasonable assumptions isolated islands are found. This provides (some) evidence for the existence of a corresponding CFT located close to these kinks.
  5. These numerical results are of physical interest and relevant to answering various experimental and methodological questions (although they do not provide conclusive answers).
  6. Useful appendices (and an ancillary .nb file) are provided so that the numerical setup can more easily be reproduced.

Weaknesses

  1. The motivation of the work could be expanded on. The paper reads as a follow up to arXiv:1904.00017 [hep-th]. That work includes more on the physical motivation for studying some of the models also studied here. It also motivates well why understanding the properties of the kinks studied in this work is important for solving some open physical questions (validity of the epsilon expansion / existence of non-perturbative fixed points / new CFTs / matching experimental results). The physical motivation is explained much more briefly in this work (probably to avoid redundancy). Perhaps it would be appropriate to offer a summary of these motivations or an explicit statement that the these can be found in the previous paper.

The included motivation, "Our motivation is to find the minimal set of conditions that allow us to isolate allowed regions in parameter space at the positions of kinks of the single correlator bounds obtained in [24].", does not sound like the strongest motivation. (See also point 1. of the requested changes.)

The authors find "such kinks to be strong indicators for the presence of actual CFTs". Therefore I would expect the motivation to be to study the characteristics of these approximate solutions in order to: 1.) Determine whether its properties are compatible with the expectations of a true CFT. 2.) To learn about physical quantities in this CFT so that it can be identified and so that these quantities can be matched with data from experiments and or alternative computational methods.

Report

This paper presents new original analytic work as well as new numerical bounds on the operator dimensions in a family of physically interesting conformal field theories. It also facilitates future conformal bootstrap research by working out the representation theory to study a wider class of theories. I recommend publication (possibly after the minor revisions / clarifications that I recommend in "Requested changes" ).

Requested changes

  1. The reason that saturation of single correlator bounds is demanded in the multi-correlator setup for some figures could be clarified.

a.) What advantage does this have over alternative methods of estimating $\Delta_S$ at the kink - such as the extremal functional method (EFM).

b.) What is the role of $\Lambda$ in these considerations?

Note that without a change in $\Lambda$ the spectrum at saturation is uniquely fixed. The EFM would instantly provide the sought for "isolated region" also in the $(\Delta_\phi,\Delta_S)$ plane. The "minimal set of conditions required in order to find an isolated region" would be only the saturation of this bound.

The benefit of the author's methods seems to be that it allows one to use information obtained in the single correlator bootstrap at large values of $\Lambda$ as input in multi-correlator bootstrap setups where similar values of $\Lambda$ are unpractical to reach. It would be useful to comment on this.

It is also interesting to look at for example Fig. 8 realizing that if $\Lambda$ were increased from 35 to 45 the island would have to shrink to a line. If this line is too expensive to compute we can instead ask what line you get when you demand $\Delta_X$ to saturate the $\Lambda=35$ bound instead of the $\Lambda=45$ one. This should not be too hard to compute (easier than the computed islands) and it would be interesting to see whether any features stand out in this line.

Related: The authors demand saturation of the $\Delta_X$ bound. However, the true solution could reasonably be expected to be located slightly below this bound. It would be nice to know how sensitive the results are to the assumed $\Delta_{X_{\text{sat}}}$. Would lowering this value slightly significantly alter the bounds.

  1. The motivation of the work could possibly be expanded on. (See Weaknesses.)

  2. The $Z\times Z$ bootstrap seems to be organized as if it were a separate section, yet it is found inside the section "3. Numerical Results". It even contains a subsection 3.5 also named "Numerical Results". I would propose that either subsections 3.3 and 3.4 are included in section 2. Or that subsection 3.3-3.5 become a new section on the $Z\times Z$ bootstrap.

  3. Do I understand correctly that the $Z \times Z$ OPE exchanges itself? If so, was it imposed that the external operator $Z$ corresponds to the lowest operator of its kind? This assumption can be made without loss of generality and could change the position of the kinks in various plots.

P.S.: Access to [29] is restricted by Kousvos, S. I am not sure whether this is intended.

---

## Round 2 · Referee Report · Anonymous (Referee 2) · 2022-4-21

Report

Warnings issued while processing user-supplied markup:

  • Inconsistency: plain/Markdown and reStructuredText syntaxes are mixed. Markdown will be used.
    Add "#coerce:reST" or "#coerce:plain" as the first line of your text to force reStructuredText or no markup.
    You may also contact the helpdesk if the formatting is incorrect and you are unable to edit your text.

The authors addressed all my questions and remarks. The changes to the paper are clear and clarify most of my questions/issues. However, the modifications raise two new questions (in order of importance):

Q1) The authors added the sentence: "We checked this by looking at the $\Delta_{SZ} \sim \Delta_Z$ exclusion bound." I think there is an important misunderstanding at play. Either on my part in the interpretation of this paragraph or on your part in understanding the assumption that should be assumed and which bounds it will afffect. I could probably have been more clear in my previous remark. Let me expand on it. In remark 4.) I pointed out that without loss of generality you can impose that the external operator $Z_{\text{ext}}$ corresponds to the lowest operator of its kind. I.e. you can impose $\Delta_Z\geq \Delta_{Z_\text{ext}}$ where $\Delta_Z$ is the dimension of the first exchanged $Z$ operator. This assumption will hold regardless of any symmetry properties of $Z_{\text{ext}}$ (i.e. regardless of whether $Z \times Z$ exchanges itself). The effect of this assumption should of course be checked in a bound that is not on $\Delta_Z$ itself, since in that bound it has no effect as a strictly stronger assumption is already being tested. Adding this assumption might very well affect the position of the kinks in for example Fig. 13 and could clear up the issue of the bound not converging to clear integer values in the large $m$.

Instead the added paragraph on page 21 seems to discuss another assumption. It is also not immediately clear how that assumption was actually imposed in practice. Assuming that the external operator $Z$ itself is exchanged would require imposing a gap above it. Are you saying you found the same for the maximal allowed value for this gap as you found for $\Delta_{Z}$ when prohibiting the exchange of $Z_{\text{ext}}$?

Q2) The change to Fig. 9. leaves the reader with a new question: Why does the allowed line at saturation of the $\Lambda=35$ bound have a gap near the kink? Is there no solution because you are imposing additional constraints compared to Fig. 8? It is somewhat unexpected that the solution disappears exactly at the kink given the results when the saturation of the $\Lambda=45$ bound is demanded. Supposedly the $\Delta_X$ bound shown in Fig. 8 is most stable under the increase of $\Lambda$ in the region around the kink (usually a kink get sharper at larger derivative order but the position of the kink itself does not change significantly). This feels a bit like counter evidence to "if $\Delta_X$ were to change a little there would be no major changes in the corresponding $\Delta_\phi-\Delta_S$ plane island." How big is the change in the $\Delta_X$ bound at the kink between $\Lambda=35$ and $\Lambda=45$ ? This amount of change clearly completely changes the nature of the bounds shown in Fig. 9.

Or viewed from another perspective: This must mean that the assumptions of Fig. 9 significantly alter the $\Delta_X$ bound exactly in the region of the kink, at least at $\Lambda=35$. I suspect that if you were to compute the green region at $\Lambda=45$ you would find the same effect, i.e. the exclusion of any solutions at all near the kink! This is a bit worrying. Strictly speaking Fig. 9 seems to be studying the empty set at least around the region of interest. In order to find a solution again you would have to instead demand saturation of the $\Delta_X$ bound found with the inclusion of the additional constraints used in Fig. 9.

Also: should we find it worrisome that your constraints seem to be most powerful close to the kink? (As also can also be seen in the "pinching" of the island saturating the $\Lambda=45$ bound.) If we view these constraints as getting rid of "fake" or "GFF"-like solution is it not weird that their effect is strongest close to the supposedly physical CFT solution?

I would leave it to the authors whether they want to address these questions but I think that clearing up the misunderstanding in Q1 could lead to a worthwhile improvement of the paper.

P.S.: in depth discussion -> in-depth discussion

  • validity: -
  • significance: -
  • originality: -
  • clarity: -
  • formatting: -
  • grammar: -

Author:  Stefanos Robert Kousvos  on 2022-05-18  [id 2493]

(in reply to Report 1 on 2022-04-21)

We thank the referee for their additional comments.
With regards to Q1:
"Are you saying you found the same for the maximal allowed value for this gap as you found for ΔZ when prohibiting the exchange of Zext?"
Yes, exactly.
We also explicitly checked that e.g. the $\Delta_{SY}$ exclusion bound for $m=1000$ and $n=4$ (seen e.g. in figure 13) remains unchanged if we impose the assumption $\Delta_{SZ} \geq 1.0$.
We propose the following modification to our text:
"We checked this by looking at the $\Delta_{SZ} \sim \Delta_{Z}$ exclusion bound..." ->
"We checked this by looking at the $\Delta_{SZ} \sim \Delta_{Z}$ exclusion bound... More explicitly, we checked that if $Z$ is exchanged in the $Z \times Z$ OPE, then the exclusion bound on the second exchanged operator in this representation is identical to the exclusion bound of the first exchanged operator in the $Z$ irrep if one assumes that the external operator is not exchanged. Additionally, we checked that for e.g. $m=1000$ and $n=4$ the corresponding $\Delta_{SY}$ exclusion bound in Fig. 14 remains unchanged even after adding the assumption $\Delta_{SZ} \geq 1.0$."
With regards to Q2:
The blue island on the left in Fig. 9 reaches, in its rightmost tip, the $\Delta_\phi$ of the kink of Fig. 9. Thus, the theory at the kink is allowed, even if only barely so. We will add a clarifying comment about this before Fig. 5: "Note that the rightmost tip of the left blue island in Fig. 9 extends to the $\Delta_\phi$ of the kink of Fig. 8. Thus, the putative theory that lives at that kink is allowed under the assumptions mentioned in the caption of Fig. 9. Given that this is a marginal case, however, further numerical work with stronger numerics and more refined methods is required to provide further clarity."
We would appreciate it if the referee could confirm to us whether they believe the above proposed modifications would sufficiently clarify the confusion in our text. We would then be happy to update the arXiv submission.

Anonymous on 2022-05-24  [id 2517]

(in reply to Stefanos Robert Kousvos on 2022-05-18 [id 2493])

Q1: The check that Fig.14 remains unchanged under the assumption $\Delta_{SZ}\geq1.0$ indicates that the assumption $\Delta_{SZ}>\Delta_{SZ_\text{ext}}$ would not make a difference in the plotted region. It is therefore reasonable not to discuss this condition in this context. However, the assumption I proposed could have been imposed without any loss of generality and it could in principle still affect the bound on $ \Delta_{BB}$ in figure 15. In fact, this is the bound where this assumption is most likely to have a significant effect since there is a kink at a large value of $\Delta_{SZ_\text{ext}}$ where this assumption would result in a significant alteration of the gap assumptions.

Q2: If the left island in Fig. 9 indeed touches the $\Delta_\phi$ of the kink this makes the gap assumptions less worrisome. However, looking at Fig 8 it seems that the kink is located around 0.5184 which is in between the two (blue) allowed regions in Fig 9 (unless I am somehow reading the figures wrongly the right most tip of the left island only reaches 0.5181). The kink is of course not that sharp and made out of multiple smaller kinks so perhaps the authors are interpreting the kink to be somewhere else. If the authors have a clear $\Delta_\phi$ value in mind perhaps a line indicating it in Fig 9 would be useful.

In my opinion, it is not unlikely that Fig 9 excludes the theory the authors attempt to study, due to the fact that the saturation of the $\Delta_X$ bound found without additional assumptions is incompatible with the additional assumptions imposed in Fig 9.

If the authors have another interpretation, for the fact that the additional conditions in Fig 9 constrain the $(\Delta_S, \Delta_\phi)$-plane the most near the location of the kink (even excluding completely the $\Delta_\phi$ values closest to the sharpest point of the kink), it would be great to include it.

Side remark: It seems that Fig 8. is only referenced to from Fig 9. and not in the main text. Is this as intended?

(Comments by Referee 1)

---

## Round 2 · Referee Report · Anonymous (Referee 1) · 2022-5-10

Report

With the changes, I think the submission now achieves the criteria for publication. With this said, there are a few spots that continue to be a little bit confusing. In the penultimate paragraph of the introduction, the authors note that “there exist experimentally relevant cases” but don’t provide any further context on this. Given that the authors already mention e.g. the cubic model, it’s worth at least referring back to that. Similarly, I think that the exact motivation for studying the X operator (that in the large-n limit it corresponds to the decoupled O(m) singlets, beyond just the fact that you need it to study S) is worth noting explicitly. I will leave it up to the authors if they would like to clarify these points, as I think in the latter case especially this is something of a squishy statement.
  • validity: -
  • significance: -
  • originality: -
  • clarity: -
  • formatting: -
  • grammar: -

Author:  Stefanos Robert Kousvos  on 2022-05-18  [id 2494]

(in reply to Report 2 on 2022-05-10)

We thank the referee for their additional comments.

We will add an additional sentence to clarify the point of experimental applications:

"...experimentally relevant cases among such models." -> "...experimentally relevant cases among such models. For example, beyond the cubic ($\mathbb{Z}_2{\!}^{3} \rtimes
S_3$), $MN_{2,2}$ ($O(2)^2\rtimes S_2$) and $MN_{2,3}$ ($O(2)^3\rtimes S_3$) models mentioned earlier, there are also the so called tetragonal theories ($D_4^n\rtimes S_n$, with $D_4$ the dihedral group of 8 elements) [a few citations will be included]."

With regards to motivations for studying the X bound, we will add a sentence explaining that the X bound is of interest specifically because it displays pronounced features/kinks. More specifically, at the end of paragraph 2 on page 3 we will add the following sentence: "We note that the $\Delta_X$ bound is of interest since it displays pronounced features/kinks in parameter space."

---

## Round 2 · Author Response

We thank both referees for their careful reading of our manuscript and constructive comments. Below we list the changes made to our draft in order to address the issues raised.

---

## Round 2 · List of Changes

Referee 1:

To address point 1, we have included a new paragraph on page 12 "Obtaining ...", we have also added in figure 9 the corresponding allowed regions that would result if one were to demand X bound saturation at $\Lambda=35$ instead of $\Lambda 45$.

With regards to point 2, we indeed didn't go into depth about the physical motivations in order to avoid redundancy, as the referee mentions. We have added an explicit reference as suggested by the referee to previous work that discusses the motivations (last sentence on first paragraph on page 3). We also changed the phrase "Our motivation..." to "Our goal..." in order to be more appropriate.

To address point 3, we have moved the $Z \times Z$ OPE analysis to the beginning of the paper.

Regarding point 4, the $Z \times Z$ OPE does indeed exchange itself. However, we found that this did not affect our plots. We added a paragraph discussing this (the last one on page 21).

Regarding the reference [29], it will become publicly available after a specific date.

Referee 2 :

To address point 1, we have added figure 7 which combines figures 6 and 5 so the reader may conveniently see their overlap. However, we opted to keep figure 4 and figure 5 separate, the reason for this is that we have separated the $\phi$ - $X$ and $\phi$ - $S$ plane islands into different sections in the text, we are thus afraid that combining these may actually cause some confusion.

To address point 2, we have moved the $Z \times Z$ OPE analysis to section 2.

---

## Round 3 · Author Response

We have incorporated a few more changes to clarify some of the additional issues raised by the referees. These are listed below.

---

## Round 3 · List of Changes

1) We added an additional sentence at the end of paragraph 2 on page 3 to clarify why we are interested in the $\Delta_X$ - $\Delta_\phi$ plane of parameter space.

2) At the top of page 4 we added some information and some additional references regarding experimentally relevant symmetries.

3) Page 3: in depth -> in-depth.

4) We added a paragraph at the bottom of page 15.

5) We expanded the paragraph on page 22 in order to specify precisely what we have explicitly checked. The main purpose of this is for the reader to know what has and has not been checked.

---

## Editorial Decision

published